# CSG-ODE: ControlSynth Graph ODE for Modeling Complex Evolution of Dynamic Graphs

**Zhiqiang Wang** [1]  **Xiaoyi Wang** [1]  **Jianqing Liang** [1]

## Abstract

Graph Neural Ordinary Differential Equations (GODE) integrate the Variational Autoencoder (VAE) framework with differential equations, effectively modeling latent space uncertainty and continuous dynamics, excelling in graph data evolution and incompleteness. However, existing GODE face challenges in capturing time-varying relationships and nonlinear node state evolution, which limits their ability to model complex dynamic graphs. To address these issues, we propose the ControlSynth Graph ODE (CSG-ODE). In the VAE encoding phase, CSG-ODE introduces an information transmission-based inter-node importance weighting mechanism, integrating it with latent correlations to guide adaptive graph convolutional recurrent networks for temporal node embedding. During decoding, CSG-ODE employs ODE to model node dynamics, capturing nonlinear evolution through sub-networks with nonlinear activations. For scenarios or prediction tasks that require stability, we extend CSG-ODE to stable CSG-ODE (SCSG-ODE) by constraining weight matrices to learnable anti-symmetric forms, theoretically ensuring enhanced stability. Experiments on traffic, motion capture, and simulated physical systems datasets demonstrate that CSG-ODE outperforms state-of-the-art GODE, while SCSG-ODE achieves both superior performance and optimal stability.

[1]Key Laboratory of Computational Intelligence and Chinese Information Processing of Ministry of Education, School of Computer and Information Technology, Shanxi University, Taiyuan 030006, Shanxi, China. Correspondence to: Jianqing Liang <liangjq@sxu.edu.cn>.

*Proceedings of the 42nd International Conference on Machine Learning*, Vancouver, Canada. PMLR 267, 2025. Copyright 2025 by the author(s).

## 1. Introduction

With the rapid development of fields such as social networks, traffic flow prediction, and financial market analysis, dynamic graph representation learning has gradually become one of the important research directions in the field of graph learning (Yang et al., 2024). A dynamic graph refers to a graph structure in which the states of nodes and edges, as well as the graph topology, evolve over time. In these systems, the interactions and relationships between nodes are time-varying, presenting both significant challenges and opportunities for model development.

In recent years, with the increasing popularity of Graph Neural Networks (GNN) (Scarselli et al., 2008), more and more dynamic graph representation learning methods have begun to leverage GNN to handle the temporal information in dynamic graphs. By introducing strategies such as temporal window mechanisms (Zhu et al., 2023) and temporal encoding (Chen et al., 2024), GNN can effectively enhance the model's ability to process dynamic graph data and capture the temporal dynamics of nodes and edges. However, many existing methods still assume that the complete state of all nodes is available at each time point. This assumption is often difficult to hold in real-world applications.

LatentODE (Rubanova et al., 2019) effectively addresses the challenge of modeling irregularly sampled time series by introducing latent ordinary differential equations (ODE) within the variational autoencoder (VAE) framework. Building upon this, Graph Neural ODE (GODE) (such as (Huang et al., 2021) and (Wen et al., 2022)) have been proposed to model the uncertainty in multidimensional time series data and capture its continuous dynamic evolution through latent space. Specifically, these methods integrate the aggregation function of GNN as part of the ODE function, effectively capturing the complex interaction information between nodes. This provides an effective tool for modeling incomplete dynamic graph data.

However, existing GODE models have limitations in capturing the time-varying relationships between nodes and the nonlinear evolution of node states. For example, in transportation systems, the road network has a significant impact on traffic flow, and its fluctuations are affected by factors

such as weather, holidays, and emergencies, leading to complex nonlinear changes. Therefore, modeling the dynamic relationships and nonlinear evolution of node states remains a key challenge in complex network modeling.

To address these issues, this paper proposes a model called ControlSynth Graph ODE (CSG-ODE) based on the VAE framework. In the encoding phase, CSG-ODE introduces node importance weights based on information propagation, overcoming the limitations of relying solely on latent space correlations between nodes. By combining node importance with latent correlations, the model effectively guides the learning of temporal node embeddings through an adaptive graph convolutional recurrent network. In the decoding phase, the model uses an ODE to model the dynamic evolution of nodes, considering not only the linear changes of nodes and their interactions with other nodes but also capturing the nonlinear evolution characteristics of node states through multiple sub-networks with nonlinear activation functions. This comprehensive approach reflects the dynamic changes in node states. To enhance stability in high-stability scenarios, we introduce the Stable CSG-ODE (SCSG-ODE) model based on CSG-ODE and theoretically demonstrate its improved stability.

Our contributions can be summarized in the following three aspects: (1) We propose the CSG-ODE model, which enables the model to better capture the complex evolution of node states through an information propagation-based inter-node importance weighting method and multiple sub-networks with nonlinear activation functions. (2) Based on CSG-ODE, we theoretically give its stabilized version SCSG-ODE. (3) The experimental results show that the proposed CSG-ODE outperforms the existing GODE model on five dynamic system datasets. And SCSG-ODE not only outperforms the existing GODE in terms of performance, but also exhibits optimal stability.

## 2. Problem Definition

Given a sequence of discrete timestamps $\mathcal{T} = \{T_1, T_2, \ldots, T_{obs}\} \in \mathbb{R}^{obs}$ with non-uniform intervals, we define the corresponding sequence of dynamic graphs $\mathcal{G}^{T_1:T_{obs}} = \{\mathcal{G}^{T_1}, \mathcal{G}^{T_2}, \ldots, \mathcal{G}^{T_{obs}}\}$, where the node connectivity remains constant and the features vary across timestamps, as illustrated in Figure 1. At timestamp $T_t$, the state of the graph is represented as $\mathcal{G}^{T_t} = (\mathcal{V}, \mathcal{E}, \mathcal{X}^{T_t})$, where $\mathcal{V}$ is the set of $N$ nodes and $\mathcal{E}$ is the set of edges. The subset of nodes with observable features at $T_t$ is denoted as $\mathcal{V}^{T_t} \subseteq \mathcal{V}$, with feature vectors $\mathcal{X}^{T_t} = \{\mathbf{x}_i^{T_t} \mid i \in \mathcal{V}^{T_t}\} \in \mathbb{R}^{|\mathcal{V}^{T_t}| \times d_1}$. Each node $i$ has an observation set $\mathcal{T}^i \subseteq \mathbb{R}^{obs\_total_i}$. Our goal is to learn low-dimensional latent representations $\mathbf{Z}^t \in \mathbb{R}^{N \times d_2}$ for each node at any timestamp $t$ from the dynamic graph sequence $\mathcal{G}^{T_1:T_{obs}}$, and use these representations to reconstruct unobserved node

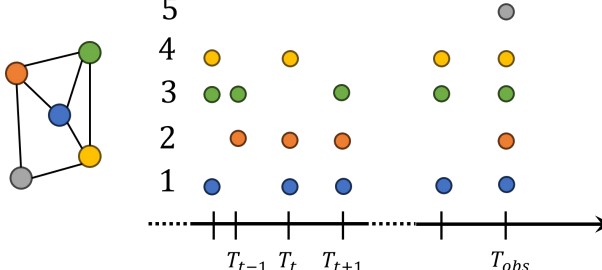

*Figure 1.* Illustration of five nodes in an dynamic graph sequence sample. The left diagram illustrates the original graph structure with fixed connections. The right diagram shows the presence or absence of data for each node at each timestamp.

attributes $\mathcal{X}^{unobs} = \{\mathbf{x}_i^t \mid i \in \mathcal{V} \backslash \mathcal{V}^t, t \in \mathcal{T} \backslash \mathcal{T}^i\}$. Additionally, we aim to predict node attributes $\mathcal{X}^{T_{obs}+1:T_{obs}+F}$ for the next $F$ timestamps.

## 3. Method

In this paper, we propose the CSG-ODE model. Following the structure of VAE, CSG-ODE consists of two main components: (1) Latent Distribution Generation: Given incomplete dynamic graph data, this component generates an approximate posterior distribution of the latent states for all nodes; (2) Latent State Generation: Based on the ODE function we designed, latent states are generated for any given timestamp by sampling the initial state for each node. The overall framework is illustrated in Figure 2.

### 3.1. Latent Distribution Generation

In this section, we describe how to learn the approximate posterior distribution of the latent states for all nodes from incomplete dynamic graph data, as shown in Figure 3.

**Enhanced Time-varying Relationship between Nodes.** Existing GODE models utilize latent representations to capture time-varying node relationships but primarily rely on latent space correlations, overlooking the impact of edges on information flow. Inspired by edge importance (Noschese & Reichel, 2024), we propose an Information Propagation-based Inter-node Importance Weight to better capture these dynamics. We compute the partial derivative of edge weights with respect to the total communicability of the graph (Benzi & Klymko, 2013). A larger derivative indicates greater contribution to information transmission, making the edge more significant. By combining inter-node importance with latent correlations, our model overcomes the limitations of existing methods that rely solely on latent space-based edge weights, this process is shown in the left part of the Figure 3.

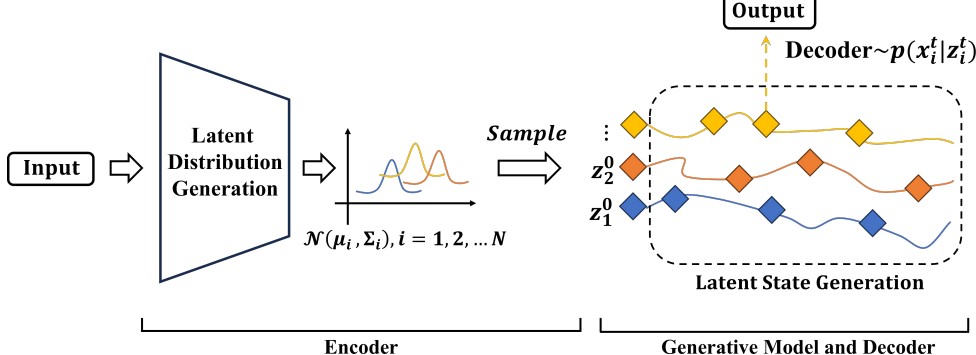

Figure 2. An overview of the proposed CSG-ODE.

Specifically, we first represent the latent correlations between nodes using a static adaptive adjacency matrix (Wu et al., 2019). This latent correlation does not require any prior knowledge and can be dynamically adjusted during the model's learning process, thereby automatically discovering the potential spatial relationships between nodes:

$$G = SoftMax(ReLU(\mathbf{E}_s\mathbf{E}_t^\top)), \tag{1}$$

where $\mathbf{E}_s, \mathbf{E}_t \in \mathbb{R}^{N \times d_1}$ represent the learnable embedding of the source and target nodes, respectively, which are randomly initialized. By multiplying $\mathbf{E}_s$ and $\mathbf{E}_t$, we obtain the latent correlation matrix $G$, where $G = [G_{ij}] \in \mathbb{R}^{N \times N}$ and $G_{ij}$ represents the edge weight or latent correlation between node $i$ and node $j$. Next, we calculate the importance weights $D = [D_{ij}] \in \mathbb{R}^{N \times N}$ between nodes:

$$L_f(G_o^\top, ee^\top) \approx \frac{\exp_0(G_o^\top + \beta ee^\top) - \exp_0(G_o^\top - \beta ee^\top)}{2\beta}, \tag{2}$$

$$D = \frac{G_o \odot L_f(G_o^\top, ee^\top)}{||L_f(G_o^\top, ee^\top)||_F}, \tag{3}$$

where $G_o \in \mathbb{R}^{N \times N}$ represents the original adjacency matrix of the graph. The vector $e = [1, 1, \ldots, 1]^\top \in \mathbb{R}^N$ and $\exp_0(G_o)$ denotes the matrix exponential minus the identity matrix, represented by the power series expansion of the exponential function ($\exp_0(G_o) = \exp(G_o) - I = G_o + \frac{G_o^2}{2!} + \frac{G_o^3}{3!} + \ldots$).

The term $L_f(G_o^\top, ee^\top)$ denotes the Fréchet derivative with respect to $G_o^\top$ and $ee^\top$, which quantifies the total transmission rate and is used to measure the contribution of each edge to the overall information flow in the graph. Equation (2) shows that this quantity is computed using a finite difference approximation, where $\beta$ denotes the step size; smaller values of $\beta$ generally yield higher approximation accuracy. We adopt the empirical formula $\beta = \frac{2}{N} \times 10^{-4}$ (Noschese & Reichel, 2024) for determining

the optimal step size. This choice is based on error analysis of the finite difference method, aiming to balance truncation and rounding errors, thereby minimizing the total numerical error. The notation $|| \cdot ||_F$ represents the Frobenius norm, with $||L_f(G_o^\top, ee^\top)||_F$ corresponding to the total transmissibility of the graph. The mixed edge weight between node $i$ and node $j$, denoted as $G_{ij}^{mix}$, is computed as follows:

$$G_{ij}^{mix} = G_{ij} + W_{ij}^1 \cdot D_{ij}, \tag{4}$$

where $W^1 \in \mathbb{R}^{N \times N}$ represents the learnable weight matrix. Therefore, by integrating both the inter-node importance and latent correlations, we address the limitations of edge weights that are solely based on latent spatial correlations between nodes. This fusion is beneficial for guiding the subsequent learning of temporal node embeddings.

**Theorem 3.1.** *For the computation of the node importance weight matrix $D \in \mathbb{R}^{N \times N}$, the time complexity is $O(N^3)$.*

**Note:** Although the subsequent Frobenius norm and associated linear operations incur an additional $O(N^2)$ cost, this is negligible in comparison to the dominant $O(N^3)$ complexity arising from matrix multiplications.

**Temporal node embedding.** The goal of Latent Distribution Generation Moudle is to learn temporal node embeddings from observable dynamic graph sequences and to approximate the posterior distribution of node latent states based on these embeddings. The temporal node embedding process is shown on the right side of the Figure 3. It captures the structural information between nodes by building a graph snapshot at each timestamp. (Zhu et al., 2016; Sankar et al., 2020). Inspired by (Luo et al., 2024c), we dynamically adjust $G^{mix}$ based on sampling density and use a mask matrix to construct graph snapshots for guiding the learning of temporal node embeddings. By adjusting edge weights based on temporal sampling density, we allocate node influence according to the differences in sampling density.

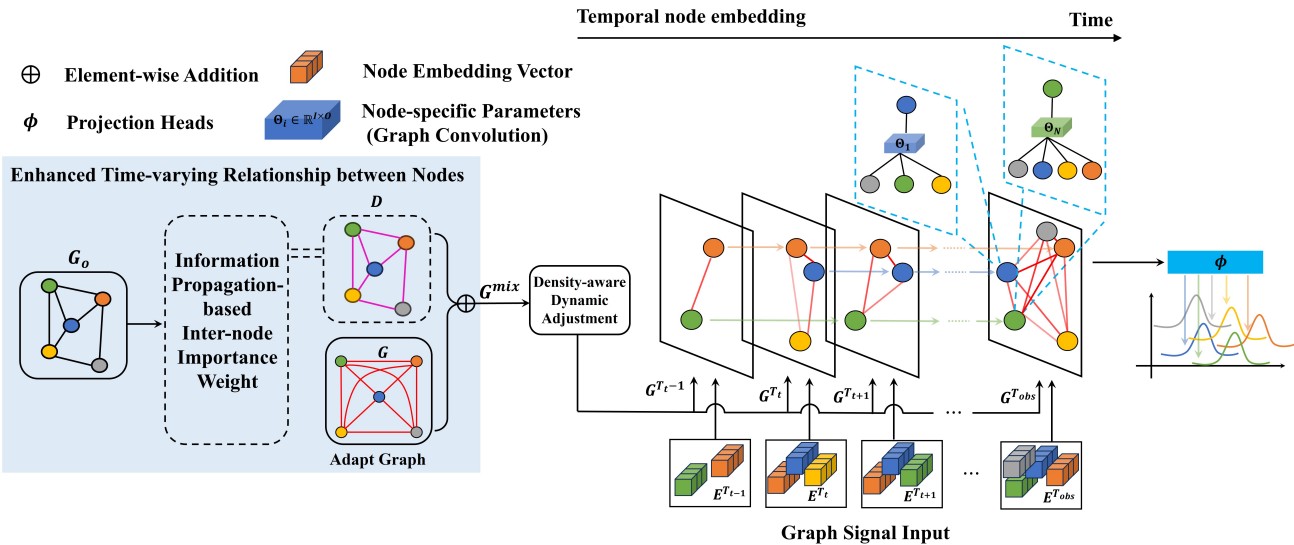

*Figure 3.* Overview of the Latent Distribution Generation in the proposed CSG-ODE.

Detailed information on the dynamic density adjustment mechanism is provided in Appendix A. Specifically, the graph snapshot at timestamp $T_t$ constructed based on the dynamic adjustment of time sampling density is as follows:

$$G_{ij}^{T_t} = G_{ij}^{mix} \cdot M_{ij}^{T_t} \times (1 - W_{ij}^2 \cdot \alpha |\sigma\left(R_i^{T_t}\right) - \sigma\left(R_j^{T_t}\right)|), \tag{5}$$

where $R_i^{T_t}$ represents the sampling density of node $i$ at timestamp $T_t$, $\sigma$ represents the activation function, $\alpha$ is the hyperparameter controlling the proportion of the time density, $W^2 \in \mathbb{R}^{N \times N}$ denotes the learnable weight matrix, and $M^{T_t} \in \mathbb{R}^{N \times N}$ represents the mask matrix at timestamp $T_t$:

$$M_{ij}^{T_t} = \begin{cases} 1, if \ both \ node \ i \ and \ j \ are \ observed \ at \ T_t, \\ 0, otherwise. \end{cases} \tag{6}$$

For the graph snapshot at timestamp $T_t$, we use a Graph Convolutional Recurrent Neural Network (GCRNN) update operator $\mathcal{F}_{\text{GCRNN}}$ that integrates graph structure and temporal dynamics:

$$\mathbf{H}^{T_t} = \mathcal{F}_{\text{GCRNN}}(G^{T_t}, \mathbf{E}^{T_t}, \mathbf{H}^{T_{t-1}}, \Theta), \tag{7}$$

where $\mathbf{H}^{T_t} \in \mathbb{R}^{N \times h}$ is the hidden state at timestamp $T_t$ and $\Theta \in \mathbb{R}^{N \times I \times O}$ denotes the graph convolution parameters, with $I$ and $O$ corresponding to the input and output feature dimensions, respectively. $\mathbf{E}^{T_t}$ represents the set of all observable embedding vectors at timestamp $T_t$, where each node's observation is obtained through a $MLP_E$:

$$\mathbf{E}_i^{T_t} = MLP_E(\mathbf{x}_i^{T_t}), T_t \in \mathcal{T}^i. \tag{8}$$

Due to the varying time evolution patterns of different nodes, the latent state distribution of each node also differs. As discussed in (Bai et al., 2020), although there may be strong spatial correlations between adjacent nodes, the dynamic nature of time series data and various factors that may influence the nodes (such as node-specific attributes) lead to diverse patterns between different nodes. Inspired by (Bai et al., 2020), we assign dedicated parameters to each node. Specifically, since the number of observable nodes varies, we further compute the average of all observable value embeddings $\mathbf{E}_i^{\text{mean}} \in \mathbb{R}^k$ for each node and use a projection function $\varphi(\cdot):\mathbb{R}^k \to \mathbb{R}^q$ represented by an $MLP$ to generate each node's embedding $\mathbf{Q}_i \in \mathbb{R}^q$:

$$\mathbf{E}_i^{mean} = \frac{1}{obs\_total_i} \sum_{j=1}^{obs\_total_i} \mathbf{E}_i^{T_{j,i}}, \tag{9}$$

$$\mathbf{Q}_i = \varphi(\mathbf{E}_i^{mean}). \tag{10}$$

The node-specific graph convolution parameters $\Theta \in \mathbb{R}^{N \times I \times O}$ are obtained by multiplying the node representation matrix $\mathbf{Q} \in \mathbb{R}^{N \times q}$ with weight matrice $W_C \in \mathbb{R}^{q \times I \times O}$, respectively:

$$\Theta = \mathbf{Q}W_C. \tag{11}$$

We obtain the approximate posterior distribution of the latent state for each node $i$ by mapping the last observed hidden state $\mathbf{H}_i = \mathbf{H}_i^{T_{obs}}$ using a function $\phi(\cdot)$, and then sample the initial latent state $z_i^0 \in \mathbb{R}^{d_2}$ for each node from this approximate posterior distribution:

$$q_\phi(z_i^0|G^{T_1:T_{obs}}) = \mathcal{N}(\mu_i, \Sigma_i), \quad where \quad \mu_i, \sigma_i = \phi(\mathbf{H}_i), \tag{12}$$

$$z_i^0 \sim p(z_i^0) \approx q_\phi(z_i^0 | \mathcal{G}^{T_1:T_{obs}}). \quad (13)$$

Detailed information on the construction method of the approximate posterior distribution and the sampling process can be found in the Appendix B.

## 3.2. Latent State Generation

In this section, we describe how to generate the latent state of each node at any given timestamp from its initial state $z_i^0$. The GODE model captures the evolution of node states over continuous time by solving an ODE, allowing for more accurate modeling of state changes. This provides a key advantage in understanding the evolution of complex dynamic graphs. Existing methods, such as (Huang et al., 2020; Luo et al., 2024a), typically model node evolution through linear dynamics and interactions with other nodes, neglecting nonlinear state evolution. For example, under adverse weather conditions like heavy rain or snow, traffic flow drops sharply, with sudden congestion emerging and worsening over time. This nonlinearity is often overlooked, limiting the model's ability to capture complex dynamics. Inspired by ControlSynth ODE (CSODE) (Mei et al., 2024), we propose a generative model that uses ODE to model both linear and nonlinear node state evolution. By incorporating multiple subnetworks with nonlinear activation functions, our model more effectively captures dynamic changes in node states.

Specifically, we introduce interactions between nodes as control information into the ODE function and use an additional control function to represent the external influence on the nodes. At the same time, we employ multiple subnetworks with nonlinear activation functions to capture the nonlinear changes in node states:

$$\begin{cases} \dot{z}_i^t = A_0 z_i^t + \sum_{j=1}^{M} A_j f_j \left( MLP_j^S(z_i^t) \right) + g(c_i^t), \\ \dot{c}_i^t = GNN(z_1^t, z_2^t, ..., z_N^t). \end{cases} \quad (14)$$

In this model, $c_i^t$ represents the interactions between nodes, i.e., the external control information of the nodes obtained through a GNN (Kipf et al., 2018). The control information directly influences the latent state of the nodes through a control function $g(\cdot)$, which is parameterized by a neural network. This mechanism helps the nodes to adapt to external factors, thereby introducing dynamic adjustments to node behavior within the model. We denote the subnetwork as $MLP_j^S$, with $f_j(\cdot)$ representing the activation function used by subnetwork $j$ (such as tanh or ReLU, etc.). In this work, we maintain that each subnetwork uses the same activation function ($f_1 = f_2 = \cdots = f_M$). The matrix $A_j$ (for $j = 1, 2, \ldots, M$) represents the learnable weight matrix, which ensures that the weight matrices appropriately combine to form the overall output of the ODE, thus enhancing the model's expressive capacity

To address the demands of high-stability scenarios, we propose the Stable CSG-ODE based on the CSG-ODE model. The SCSG-ODE is characterized by the following ODE form:

$$\dot{z}_i^t = A z_i^t + \sum_{j=1}^{M} A f_j \left( MLP_j^S(z_i^t) \right) + g(c_i^t), \quad (15)$$

where $A$ is a learnable antisymmetric matrix.

*Remark* 3.2. For any matrix $A$, we can control it to be an antisymmetric matrix by using $A - A^\top$.

**Stability Analysis:** Consider the ODE defined by Equation (15). Under certain conditions, we can prove that the system is stable. Specifically, when solving this ODE, regardless of the initial values (initial states) sampled from the approximate posterior distribution, the state evolution described by the equation remains consistent and is not significantly affected by the initial values. This demonstrates that our model is insensitive to the initial conditions.

**Definition 3.3.** The initial conditions are insensitive: Even when the initial conditions of the equation are subject to minor perturbations, the solution will remain within a finite and stable range.

When the depth of each subnetwork is set to 1, the subnetwork $MLP_j^S$ can be equivalently represented by a weight matrix $\widetilde{W}_j \in \mathbb{R}^{d_2 \times d_2}$. Under this setting, we present the stability theorem of Equation (15) as follows:

**Theorem 3.4.** *Assuming that the control information $c_i^t$ does not depend on $z_i^t$, and that for any $i, j \in \{1, 2, ..., M\}$, the condition $\widetilde{W}_j \widetilde{W}_i = \widetilde{W}_i \widetilde{W}_j$ holds. Then, the Equation (15) is stable.*

Based on Theorem 3.4, we can conclude that the stability of the system ensures that when different initial latent states $z_i^0$ are sampled from the approximate posterior distribution, the solution of Equation (15) will not experience drastic fluctuations. Even in the presence of small perturbations, the solution remains stable within a finite range, preventing significant oscillations. Theorem 3.4 provides theoretical guarantees for the stability of the model. A detailed proof of this theorem can be found in the Appendix C.

By solving the ODE function, we obtain the latent state at the desired time step:

$$z_i^0, ..., z_i^{pred} =$$
$$ODESolve(Equation\ (14), [z_1^0, ..., z_N^0], (t_0, ..., t_{pred})). \quad (16)$$

Given the initial latent states of each node $z_1^0, z_2^0, \ldots, z_N^0$, analytical solutions for ODE are typically unavailable. Therefore, numerical methods, such as Euler's method, are employed to approximate the solution. A numerical ODE

solver can compute the approximate value of the node state $z_i^t$ at any time step $t$, assisting in simulating the system's behavior over time.

Finally, a decoder, represented by a neural network, is used to decode the probability $p(x_i^t | z_i^t)$ and recover the node features $x_i^t$. Under the VAE framework, all modules are jointly trained by maximizing the evidence lower bound (ELBO) of the likelihood function, as shown below:

$$\mathcal{L}_{ELBO} = \mathbb{E}_{\mathbf{Z}^0 \sim \prod_{i=1}^{N} q_\phi(z_i^0 | \mathcal{G}^{T_1 : T_{obs}})}[log_p(\mathcal{X}^{t_0 : t_{pred}})] -$$
$$KL[\prod_{i=1}^{N} q_\phi(z_i^0 | \mathcal{G}^{T_1 : T_{obs}}) || p(\mathbf{Z}^0)],$$
$$(17)$$

where $p(\mathbf{Z}^0) = \prod_{i=1}^{N} p(z_i^0)$ and $p(z_i^0)$ is a standard normal distribution. The training process of this model is detailed in Algorithm 1 in the Appendix I.

## 4. Experiments

### 4.1. Datasets

We evaluate our model on five diverse datasets: two synthetic datasets, Springs (Kipf et al., 2018) and Charged (Kipf et al., 2018), and three real-world datasets—CMU motion capture data (walk capture from subject 35 and jump capture from subject 118) (CMU, 2003), and the PEMS08 traffic flow dataset (Song et al., 2020). Further details are provided in Appendix D.

### 4.2. Baselines

To evaluate the effectiveness of the proposed CSG-ODE in the task, we designed a comparative study with several existing methods, including four baseline models: Latent-ODE (Rubanova et al., 2019), Edge-GNN (Gong & Cheng, 2019), Weight-Decay (Cao et al., 2018), and LG-ODE (Huang et al., 2020), as well as two advanced model variants: NRI+RNN (Huang et al., 2020) and LG-CSODE. Detailed descriptions of these models can be found in the Appendix F.

### 4.3. Experiment Setup

We evaluate our model on two tasks(Rubanova et al., 2019):

- **Interpolation Task:** Given a dynamic graph with evolving nodes over the time range $(t_0, t_n)$, we observe their state features at specific time points, with partial observations due to practical constraints. The task is to predict the complete dynamic trajectory of nodes over the time range $\{t_0, t_1, \ldots, t_n\}$, based on a subset of observed data. The subsampling ratio is set to 40%, 60%, and 80%, and the observations are independent across nodes. The objective is to predict the missing feature values for all time points. We eval-

uate model performance using the Mean Squared Error (MSE) metric (Huang et al., 2020; Luo et al., 2024a; Huang et al., 2024), comparing the reconstructed trajectory with the true trajectory.

- **Extrapolation Task:** We split the time range into two segments: $(t_0, t_{n1})$ and $(t_{n1}, t_n)$. During training, the model reconstructs the trajectory for the second segment $(t_{n1}, t_n)$ using observations from the first segment. In testing, the model predicts the trajectory for the future time range $(t_n, t_{n2})$ based on observations from $(t_0, t_n)$. The model is conditioned on partial observations (40%, 60%, and 80%) from the first segment, and the reconstructed trajectory is evaluated using the MSE metric.

Further details on the experimental setup and hyperparameters can be found in Appendix E.

### 4.4. Results

Table 1 presents the MSE for interpolation tasks across different datasets and methods. Latent-ODE, which does not consider the interaction information between nodes, exhibits poor performance. Edge-GNN, although propagating temporal dynamic information in the graph, fails to capture complex temporal dependencies, resulting in suboptimal performance. Weight-Decay does not fully account for the dynamic interactions between nodes and only performs a simple parameterization of time intervals, leading to unsatisfactory results. LG-ODE leverages neural ODE to infer complex latent dynamics but still falls short in modeling the temporal relationships and nonlinear evolution between nodes, limiting its performance. Although LG-CSODE employs our designed ODE function for modeling, it relies solely on latent correlations to model node relationships during encoding, which negatively impacts performance. As the observation ratio increases, the reconstruction loss for all models decreases, as expected, because more observed data helps the model more accurately reconstruct the unobserved portions.

Table 2 shows the MSE for extrapolation tasks, which is higher compared to interpolation tasks, as predicting future trajectories is more challenging than reconstructing missing data. Similar to interpolation tasks, the prediction errors for all models decrease gradually as the observation ratio increases. Our model demonstrates significant performance improvements across multiple datasets and settings, further validating the effectiveness of the proposed design. The results are visualized in the Appendix G.

Furthermore, we analyzed the potential sources of error in CSG-ODE. The model assumes that the underlying interaction graph remains static over time; however, this assumption does not fully hold in real physical systems, which may

*Table 1.* Mean Squared Error (MSE$\times 10^{-2}$) on Interpolation task.

| Dataset | Springs | | | Charged | | | Motion-walk | | | Motion-jump | | | PEMS08 | | |
|---|---|---|---|---|---|---|---|---|---|---|---|---|---|---|---|
| ration | 40% | 60% | 80% | 40% | 60% | 80% | 40% | 60% | 80% | 40% | 60% | 80% | 40% | 60% | 80% |
| Latent-ODE | 0.4729 | 0.4207 | 0.3757 | 1.0185 | 1.0352 | 0.7411 | 0.1405 | 0.0799 | 0.0766 | 0.1598 | 0.1688 | 0.1028 | 0.4855 | 0.4710 | 0.4672 |
| Weight-Decay | 0.9127 | 0.9504 | 0.9416 | 2.4532 | 2.0855 | 1.3690 | 0.3465 | 0.3314 | 0.3139 | 0.3940 | 0.7005 | 0.4820 | 1.0356 | 1.0641 | 1.1443 |
| Edge-GNN | 1.0488 | 1.0681 | 0.7171 | 1.3635 | 1.4446 | 1.0158 | 0.5044 | 0.4414 | 0.4305 | 0.5736 | 0.5571 | 0.5274 | 1.1901 | 1.1959 | 1.0826 |
| NRI+RNN | 0.4098 | 0.3382 | 0.3107 | 1.2010 | 1.0784 | 0.9201 | 0.1065 | 0.0893 | 0.1211 | 0.1212 | 0.1287 | 0.1185 | 0.4651 | 0.3787 | 0.4691 |
| LG-ODE | 0.2628 | 0.2648 | 0.2313 | 0.7971 | 0.7656 | 0.7157 | 0.0823 | 0.0475 | 0.0795 | 0.0936 | 0.1004 | 0.0974 | 0.2982 | 0.2965 | 0.3492 |
| LG-CSODE | 0.2158 | 0.1666 | 0.1937 | **0.7794** | 0.7556 | 0.7229 | 0.0461 | 0.0426 | 0.0435 | 0.0665 | 0.0648 | 0.0649 | 0.3214 | 0.2890 | 0.3525 |
| **Ours** | **0.1550** | **0.1440** | **0.1386** | 0.7947 | **0.7169** | **0.7099** | **0.0439** | **0.0406** | **0.0400** | **0.0426** | **0.0414** | **0.0405** | **0.2526** | **0.2827** | **0.3360** |

*Table 2.* Mean Squared Error (MSE$\times 10^{-2}$) on Extrapolation task.

| Dataset | Springs | | | Charged | | | Motion-walk | | | Motion-jump | | | PEMS08 | | |
|---|---|---|---|---|---|---|---|---|---|---|---|---|---|---|---|
| ration | 40% | 60% | 80% | 40% | 60% | 80% | 40% | 60% | 80% | 40% | 60% | 80% | 40% | 60% | 80% |
| Latent-ODE | 5.5751 | 3.7938 | 3.8878 | 11.7554 | 10.7649 | 18.8735 | 0.7222 | 1.0262 | 0.9042 | 0.5166 | 0.5331 | 0.5947 | 6.2301 | 4.4778 | 3.8409 |
| Weight-Decay | 5.1282 | 5.1279 | 4.8347 | 8.2000 | 7.6614 | 7.3471 | 5.0175 | 4.8270 | 4.6517 | 3.5887 | 2.9780 | 3.0594 | 5.7307 | 6.0525 | 4.7664 |
| Edge-GNN | 5.0331 | 4.3959 | 2.9056 | 7.9716 | 7.6917 | 7.1133 | 3.9712 | 4.9322 | 3.3552 | 2.8404 | 2.7429 | 2.2067 | 5.6244 | 5.1885 | 3.8706 |
| NRI+RNN | 2.2191 | 2.1437 | 2.2998 | 6.2108 | 5.8829 | 6.1291 | 1.0565 | 1.0636 | 0.8843 | 0.7556 | 0.6562 | 0.5816 | 2.4798 | 2.5302 | 2.2721 |
| LG-ODE | 1.4861 | 1.6151 | 1.5427 | 5.6522 | 5.4162 | 5.7353 | 0.3835 | 0.4391 | 0.4084 | 0.2743 | 0.2709 | 0.2686 | 1.7726 | 1.9063 | 1.5241 |
| LG-CSODE | 1.3508 | 1.2975 | 1.2750 | 5.8291 | 5.5811 | 5.6171 | 0.1853 | 0.1583 | 0.1638 | 0.2536 | 0.2837 | 0.3242 | 2.4391 | 3.5231 | 3.4111 |
| **Ours** | **1.3495** | **1.2969** | **1.2691** | **5.5086** | **4.7690** | **4.4966** | **0.1791** | **0.1539** | **0.1593** | **0.1393** | **0.1290** | **0.1248** | **1.6607** | **1.7436** | **1.4937** |

introduce a certain degree of bias. Numerical errors inherent in the ODE solving process are also unavoidable. In addition, real-world data are often influenced by complex factors such as environmental changes, noise, and latent unmodeled dynamics, none of which are considered in the current model, potentially further exacerbating error accumulation.

### 4.5. Ablation Study

To further analyze the components of the model, we conduct an ablation study by considering four model variants. First, due to the differing time patterns across nodes, simply sharing a single parameter space for all nodes is insufficient to capture these differentiated time dependencies. Therefore, we adjust the parameter space for each node according to its representation to accommodate its respective time pattern. We compare the learnable adaptive node representation matrix $\mathbf{Q} \in \mathbb{R}^{V \times q}$ with the handcrafted node representation matrix, where the former is denoted as Ours-AQ. Additionally, we remove the node importance weights based on information propagation, resulting in the variant Ours-no EI. Second, in terms of the control term in the neural differential equation, we remove the control function $g(\cdot)$, which influences the node states, effectively using only the interaction information between nodes. This variant is denoted as Ours-no $g(\cdot)$. Finally, we remove the nonlinear term in the neural ODE equation, retaining only the control information and linear terms; this variant is referred to as Ours-no NI. Through these variants, we aim to gain a deeper understanding of the role of each component in the model and its contribution to the overall performance.

The results in Table 3 and Table 4 reveal several key findings: (1) Using the mean of node observations to construct node representations yields better interpretability and performance than adaptive node embeddings. (2) Removing the information transmission-based inter-node importance weighting causes a significant performance drop, as this weighting encodes both the graph topology and the role of edges in global information propagation. Integrating this weighting with latent pairwise correlations enhances the model's ability to capture time-varying inter-node relationships. (3) Excluding the control function $g(\cdot)$ degrades performance, since $g(\cdot)$ bridges external control inputs and internal dynamics; its absence reduces the node's sensitivity to external influences. (4) Removing the nonlinear term weakens the model's capacity to capture complex dynamics, as real-world systems are inherently nonlinear; incorporating nonlinearity improves expressiveness and adaptability.

### 4.6. Parameter Sensitivity

To investigate the impact of the hyperparameter $\alpha$, which controls the proportion of time density, we conducted experiments using walk capture data from Subject 35. We varied $\alpha$ from the set $\{0, 0.1, 0.2, 0.3, 0.4, 0.5, 0.6, 0.7, 0.8, 0.9, 1\}$ and analyzed its effect across two tasks. As shown in Figure 4, for the interpolation task, changes in $\alpha$ had minimal impact, and the model's performance remained stable. This suggests that, regardless of sampling density, the model can effectively infer missing data based on existing trends. For the extrapolation task, we observed that when the observation ratio was low, changes in $\alpha$ had little effect. However, as the observation ratio increased, performance improved with higher sampling density, as the increased density better captured the data dependencies. Accordingly, we selected $\alpha = 0.5$ for the majority of experiments, since the

Table 3. Ablation study on all datasets (MSE $\times 10^{-2}$) for the Interpolation task.

| Dataset | Springs | | | Charged | | | Motion-walk | | | Motion-jump | | | PEMS08 | | |
|---|---|---|---|---|---|---|---|---|---|---|---|---|---|---|---|
| ration | 40% | 60% | 80% | 40% | 60% | 80% | 40% | 60% | 80% | 40% | 60% | 80% | 40% | 60% | 80% |
| CSG-ODE AQ | **0.1520** | 0.1526 | 0.1463 | 1.8275 | 1.0288 | 0.8397 | 0.0674 | 0.0614 | 0.0618 | 0.0880 | 0.0691 | 0.0664 | 0.7949 | 0.6989 | 0.6058 |
| CSG-ODE no EI | 0.1841 | 0.1712 | 0.1577 | 0.8005 | 0.7323 | 0.7855 | 0.0470 | 0.0435 | 0.0411 | 0.0440 | 0.0429 | 0.0413 | 0.2644 | 0.2531 | 0.4137 |
| CSG-ODE no $g(\cdot)$ | 0.1655 | 0.1918 | 0.1417 | 0.8236 | 0.6531 | 0.7422 | 0.0480 | 0.0440 | 0.0433 | 0.0432 | 0.0449 | 0.0436 | 0.2595 | **0.2466** | 0.4857 |
| CSG-ODE no NI | 0.1913 | 0.1592 | 0.1507 | 0.7950 | 0.7775 | 0.7768 | 0.0456 | 0.0426 | 0.0478 | 0.0430 | **0.0409** | 0.0443 | 0.2681 | 0.2811 | 0.3744 |
| CSG-ODE | 0.1550 | **0.1440** | **0.1386** | **0.7947** | **0.7169** | **0.7099** | **0.0439** | **0.0406** | **0.0400** | **0.0426** | 0.0414 | **0.0405** | **0.2526** | 0.2602 | **0.3360** |

Table 4. Ablation study on all datasets (MSE $\times 10^{-2}$) for the Extrapolation task.

| Dataset | Springs | | | Charged | | | Motion-walk | | | Motion-jump | | | PEMS08 | | |
|---|---|---|---|---|---|---|---|---|---|---|---|---|---|---|---|
| ration | 40% | 60% | 80% | 40% | 60% | 80% | 40% | 60% | 80% | 40% | 60% | 80% | 40% | 60% | 80% |
| CSG-ODE AQ | 4.4046 | 3.0096 | 2.0561 | 11.8165 | 8.3525 | 6.5031 | 0.2337 | 0.1840 | 0.1695 | 0.2620 | 0.2075 | 0.1978 | 2.8898 | 2.6808 | 2.5539 |
| CSG-ODE no EI | 2.4394 | 2.0790 | 1.6435 | 5.6443 | 5.0040 | 4.5996 | **0.1784** | 0.1549 | **0.1212** | 0.1413 | 0.1298 | 0.1253 | 1.9424 | 1.7881 | 1.7049 |
| CSG-ODE no $g(\cdot)$ | 2.4239 | 2.0039 | 1.4903 | 5.6228 | 5.0025 | 4.6865 | 0.1849 | 0.1545 | 0.1494 | 0.1424 | 0.1311 | 0.1264 | 1.9448 | **1.6178** | 2.2629 |
| CSG-ODE no NI | 1.5368 | 1.5077 | 1.4983 | 5.5850 | 5.0550 | 4.5048 | 0.1954 | 0.1635 | 0.1601 | 0.1405 | 0.1293 | 0.1261 | 1.7763 | 1.8778 | 1.5109 |
| CSG-ODE | **1.3495** | **1.2969** | **1.2691** | **5.5086** | **4.7690** | **4.4966** | 0.1791 | **0.1539** | 0.1593 | **0.1393** | **0.1290** | **0.1248** | **1.7726** | 1.7436 | **1.4937** |

results tend to stabilize and achieve a local optimum around $\alpha = 0.5$ in the extrapolation experiments.

### 4.7. SCSG-ODE Performance

We performed a comparison experiment of SCSG-ODE with walk capture data from subject 35 with baselines. The Table 5 indicate that CSG-ODE outperforms all other models, while SCSG-ODE achieves near-optimal performance, a result that is reasonable. Compared to CSG-ODE, SCSG-ODE fixes the learnable weight matrix to the same skew-symmetric matrix, while different subnetworks simulate the nonlinear evolution of the nodes. Using distinct weight matrices ensures that these subnetworks, when appropriately combined, produce the overall output of the model, thus enhancing the model's expressive power. Although CSG-ODE outperforms SCSG-ODE, we have theoretically demonstrated that by controlling the weight matrices, this design significantly improves the model's stability. Moreover, relative to LG-ODE, NRI+RNN, Edge-GNN, Weight-Decay, and Latent-ODE models, the performance of SCSG-ODE is optimal. To verify the correctness of the theoretical derivations, we conducted additional experiments on the performance of CSG-ODE and SCSG-ODE in the extrapolation task. Specifically, we conducted 5 rounds of experiments for each model with different subsampling ratios on walk motion capture data from subject 35 and calculated their means and standard deviations. The experimental results are shown in the Table 6. It is worth noting that the standard deviation of SCSG-ODE is always about half of that of CSG-ODE under all subsampling ratios, which indicates that SCSG-ODE significantly improves the stability of the model under the same experimental setup, and thus verifies the correctness of our theoretical derivations. Therefore, the SCSG-ODE is an ideal choice when we face the demands of scenarios that require high stability.

Table 5. Performance comparison of SCSG-ODE, CSG-ODE, LG-ODE, NRI+RNN, Edge-GNN, Weight-Decay and Latent-ODE on motion-walk datasets. The best and second best results are highlighted in blue and brown, respectively.

| Task | Interpolation | | | Extrapolation | | |
|---|---|---|---|---|---|---|
| ratio | 40% | 60% | 80% | 40% | 60% | 80% |
| Latent-ODE | 0.1405 | 0.0799 | 0.0766 | 0.7222 | 1.0262 | 0.9042 |
| Weight-Decay | 0.3465 | 0.3314 | 0.3139 | 5.0175 | 4.8270 | 4.6517 |
| Edge-GNN | 0.5044 | 0.4414 | 0.4305 | 3.9712 | 4.9322 | 3.3552 |
| NRI+RNN | 0.1065 | 0.0893 | 0.1211 | 1.0565 | 1.0636 | 0.8843 |
| LG-ODE | 0.0823 | 0.0475 | 0.0795 | 0.3835 | 0.4391 | 0.4084 |
| CSG-ODE | 0.0439 | 0.0406 | 0.0400 | 0.1791 | 0.1539 | 0.1593 |
| SCSG-ODE | 0.0435 | 0.0410 | 0.0438 | 0.2447 | 0.2035 | 0.1883 |

Table 6. Comparison of the stability performance of SCSG-ODE and CSG-ODE on motion-walk.

| ratio | 40% | 60% | 80% |
|---|---|---|---|
| CSG-ODE | $0.1883 \pm 0.0092$ | $0.1676 \pm 0.0109$ | $0.1524 \pm 0.0097$ |
| SCSG-ODE | $0.2304 \pm 0.0056$ | $0.1978 \pm 0.0050$ | $0.1787 \pm 0.0043$ |

## 5. Related Work

### 5.1. GNN for Dynamic Graph Representation

GNN (Scarselli et al., 2008) have become a powerful tool for dynamic graph representation learning. In dynamic graphs, time affects not only node features and edge weights but also the structure of the graph itself. Recent GNN-based models (Wu et al., 2019; Hajiramezanali et al., 2019; Rossi et al., 2020) focus on capturing temporal changes in node features. These models have been widely applied in areas such as traffic prediction (Zhao et al., 2019; Guo et al., 2020) and healthcare (Gao et al., 2021; Wang & Jin, 2025).However, most of these models assume complete, synchronized observations with uniform time intervals. This assumption limits their effectiveness in handling incomplete data, particularly when time intervals are irregular. To address this,

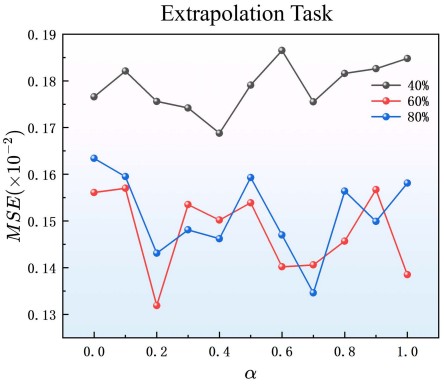 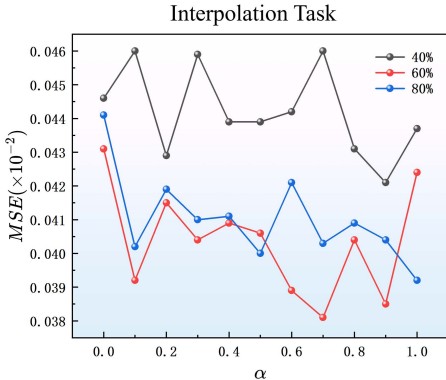

*Figure 4.* Comparison of the effect of different $\alpha$ values on model performance in two tasks.

we introduce a VAE-based framework, using a GNN encoder to infer an approximate posterior distribution of node latent states from irregularly sampled partial data. We also integrate a neural ODE to model the dynamic evolution of node states, enabling the generation of latent node states at any timestamp and overcoming the limitations of existing methods in incomplete data scenarios.

## 5.2. NODE for Irregularly Sampled Multivariate Time Series Modeling

Modeling irregularly sampled multivariate time series is challenging due to uneven time intervals and incomplete observations. Neural Ordinary Differential Equations (NODE) (Chen et al., 2018) combine neural networks and ODE to model dynamic systems and have proven effective for time series data. Several variants have been proposed (Rubanova et al., 2019; Kidger et al., 2020; Mei et al., 2024), with some addressing the GNN over-smoothing problem and extending its use in multivariate time series modeling through GNN or message-passing mechanisms (Deng et al., 2019; Poli et al., 2019; Xhonneux et al., 2020).Recently, GODE has been integrated into the VAE framework to model irregularly sampled multivariate time series (Huang et al., 2021; Wen et al., 2022; Yıldız et al., 2022; Luo et al., 2024b; Gravina et al., 2024). LG-ODE (Huang et al., 2020) employs unsupervised learning to infer initial states from irregular observations and uses NODE for continuous-time latent dynamics. (Yıldız et al., 2022) accurately decompose the independent dynamics of individual objects from their interactions and infer the independent dynamics and its interaction with reliable uncertainty estimates using ODE for underlying Gaussian processes. PGODE (Luo et al., 2024a) incorporates latent representations into the graph ODE, enhancing the model's expressiveness by identifying interaction prototypes. However, existing GODE models mainly capture latent correla-

tions between nodes but overlook nonlinear changes in node states. In contrast, our model addresses this limitation by introducing node importance weights based on information propagation. During the decoding phase, we use multiple subnetworks with nonlinear activation functions to capture the nonlinear evolution of node states, providing a more comprehensive representation of their dynamic behavior.

## 6. Conclusion

The CSG-ODE model proposed in this paper effectively addresses the limitations of existing GODE models in modeling time-varying relationships between nodes by introducing node importance weights based on information propagation. Additionally, the model utilizes multiple sub-networks with nonlinear activation functions to capture the nonlinear evolution of node states, further enhancing its expressive power. To improve the model's stability, we introduce an extended version, SCSG-ODE, and theoretically demonstrate that this extension significantly enhances stability. Experimental results show that CSG-ODE outperforms existing GODE models across multiple datasets. Although CSG-ODE outperforms SCSG-ODE in terms of performance, the latter demonstrates a more pronounced advantage in stability. Future research could focus on further optimizing the computational efficiency and generalization ability of both CSG-ODE and SCSG-ODE for large-scale and more complex graph data.

## Acknowledgement

This work is supported by the National Natural Science Foundation of China (Nos. 62272285, 62376142, U21A20473).

## Impact Statement

This paper presents work whose goal is to advance the field of Machine Learning. There are many potential societal consequences of our work, none which we feel must be specifically highlighted here.

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

## A. Detail of Density-aware Dynamic Adjustment

By adjusting the edge weights between nodes based on temporal sampling density (Luo et al., 2024c), we can reasonably allocate the influence of each node in the graph according to the differences in sampling density. Nodes with lower sampling density may have less accurate or incomplete information, so their edge weights with other nodes are reduced to avoid excessive interference from inaccurate state data during node evolution. In contrast, nodes with higher sampling density have more accurate information, enabling effective information sharing with other nodes, thus their edge weights are larger, ensuring their influence in the graph. The calculation formula for the sampling density $R_i^{T_t}$ of node $i$ at each observable timestamp $T_t$ is as follows:

$$R_i^{T_t} = \begin{cases} \frac{(T_{t,i} - T_{t-1,i}) + (T_{t+1,i} - T_{t,i})}{2} & if\ both\ T_{t-1,i}\ and\ T_{t+1,i}\ exist, \\ T_{t,i} - T_{t-1,i} & if\ only\ T_{t-1,i}\ exist, \\ T_{t+1,i} - T_{t,i} & if\ only\ T_{t+1,i}\ exist, \\ \frac{T_{obs} - T_1}{2} & if\ neither\ T_{t-1,i}\ nor\ T_{t+1,i}\ exist. \end{cases}$$

The sampling density is determined by calculating the average time interval between each observation point and its preceding and succeeding observation points. For timestamp $T_t$, if node $i$ has no preceding or succeeding observation points at this timestamp, it indicates that this is the only observation point for node $i$. In this case, its sampling density is set to half of the maximum observation time span.

## B. Detail of Distribution Generation and Sampling

To obtain the approximate posterior distribution of the latent state for each node, we input the hidden state $\mathbf{H}_i \in \mathbb{R}^h$ updated at the last observed timestamp of node $i$ into a MLP $\phi(\cdot) : \mathbb{R}^h \to \mathbb{R}^{2d_2}$, which outputs a distribution vector $\mathbf{H}^D \in \mathbb{R}^{2d_2}$:

$$\mathbf{H}^D = \phi(\mathbf{H}_i).$$

Next, we split this vector into two parts, with the first $d_2$-dimensional vector representing the mean vector $\mu_i = [\mu_{i,1}, \mu_{i,2}, ..., \mu_{i,d_2}] \in \mathbb{R}^{d_2}$ of node $i$, and the second $d_2$-dimensional vector representing the variance vector $\sigma_i = [\sigma_{i,1}, \sigma_{i,2}, ..., \sigma_{i,d_2}] \in \mathbb{R}^{d_2}$ of node $i$. Notably, the approximate posterior distribution of the latent state generated by the node is a $d_2$-dimensional Gaussian distribution $\mathcal{N}(\mu_i, \Sigma_i)$, with the assumption that the distributions across dimensions are independent. Thus, the covariance matrix $\Sigma_i$ is represented as a diagonal matrix $diag(\sigma_{i,1}, \sigma_{i,2}, ..., \sigma_{i,d_2})$, where the diagonal elements correspond to the variance in each dimension of the variance vector, and each element of $\mu_i$ and $\sigma_i$ corresponds to the mean and variance of the Gaussian distribution in each dimension, respectively.

We apply the Reparameterization Trick (Kingma et al., 2013) to sample the initial latent state $z_i^0 \in \mathbb{R}^{d_2}$ for each node from this approximate posterior distribution $\mathcal{N}(\mu_i, \Sigma_i)$. Specifically, when sampling from a multivariate Gaussian distribution where each dimension's distribution is independent, we first sample from each univariate Gaussian distribution, then combine them into a $d_2$-dimensional vector:

$$z_i^0 = (\mu_{i,1} + \sigma_{i,1} * \varepsilon_1, \mu_{i,2} + \sigma_{i,2} * \varepsilon_2, ..., \mu_{i,d_2} + \sigma_{i,d_2} * \varepsilon_{d_2}),$$

where $\varepsilon \sim \mathcal{N}(0, 1)$ is sampled from the standard normal distribution.

## C. Proof of Theorem 3.4

**Lemma C.1.** *(Haber & Ruthotto, 2017; Gravina et al., 2024)The equation is stable if and only if the real part of all eigenvalues of the Jacobian matrix $J(t)$ of the equation is zero, i.e., $Re(\lambda_i(J(t))) = 0, \forall i = 1, \ldots, d_2$.*

*Proof.* Since we set the depth of each subnetwork to 1 and the subnetwork $\mathrm{MLP}_j^S$ is represented as the weight matrix $\widetilde{W}_j$, the equation can be simplified to the following form:

$$\dot{z}_i^t = Az_i^t + \sum_{j=1}^{M} Af_j(\widetilde{W}_j(z_i^t)) + g(c_i^t).$$

The eigenvalues of the Jacobian matrix are key factors in controlling the sensitivity of the solution to initial conditions. The Jacobian matrix describes how the system state responds to small perturbations, and its eigenvalues determine the rate of

expansion or contraction of the system in different directions.

$$J(t) = (I + \sum_{j=1}^{M} \widetilde{W}_j f'(z_i^t))A.$$

Let $P = (I + \sum_{j=1}^{M} \widetilde{W}_j f'(z_i^t))$. In the space of non-diagonal real $d_2 \times d_2$ matrices, the set of non-diagonalizable matrices (in the complex field) has zero measure (Bhatia, 2013). Therefore, under random initialization, the matrix $\widetilde{W}_j$ is diagonalizable with probability 1,i.e., $\widetilde{W}_j = V_j \Lambda_j V_j^{-1}$, where $\Lambda_j = diag(\lambda_{j_1}, \ldots, \lambda_{j_{d_2}}) \in \mathbb{C}^{d_2 \times d_2}$. By assumption, for any $i, j \in \{1, 2, ..., M\}$, all $\widetilde{W}_j \widetilde{W}_i = \widetilde{W}_i \widetilde{W}_j$ holds, then all matrices $\widetilde{W}_j$ can be diagonalized simultaneously. Hence, by similarity, we can rewrite it as:

$$P = (I + \sum_{j=1}^{M} \Lambda_j f'(z_i^t)).$$

Moreover, since the derivative of the activation function $\tanh$ is non-negative, $P$ is clearly invertible. Therefore, the Jacobian matrix is the result of the matrix multiplication between an invertible diagonal matrix and the weight matrix, i.e.,

$$J(t) = PA.$$

Let $\lambda$ be an eigenvalue of $PA$ with the corresponding eigenvector $\widetilde{v}$. We haves

$$PA\widetilde{v} = \lambda\widetilde{v},$$

$$A\widetilde{v} = \lambda P^{-1}\widetilde{v},$$

$$\widetilde{v}^* A\widetilde{v} = \lambda(\widetilde{v}^* P^{-1}\widetilde{v}).$$

where * denotes the conjugate transpose, and it is important to note that $\widetilde{v}^* P^{-1}\widetilde{v}$ is a real number. Since $A$ is a anti-symmetric matrix over the complex field, we have $A^* = A^\top = -A$,and thus

$$(\widetilde{v}^* A\widetilde{v})^* = \widetilde{v}^* A^* \widetilde{v} = -\widetilde{v}^* A\widetilde{v}.$$

This implies that $\widetilde{v}^* A\widetilde{v}$ must be purely imaginary, because the conjugate transpose of a real number equals its negative, which only holds when the number is purely imaginary. Since $\widetilde{v}^* A\widetilde{v}$ is purely imaginary, it follows that $\lambda$ must be purely imaginary, i.e, $\text{Re}(\lambda_i(J(t))) = 0$ for all $i = 1, \ldots, d_2$. By Lemma C.1, this proves that the equation is stable. $\qquad\square$

## D. Dataset Details

### D.1. Simulated Datasets

Following the method in (Kipf et al., 2018), we constructed two simulated datasets: Spring and Charged. In the spring system, particles are interconnected by springs, and their interactions follow Hooke's law. Each particle interacts with others with equal probability. In the charged particle system, particles carry electric charges and interact via electromagnetic forces, which can be either attractive or repulsive, each occurring with equal probability. Each dataset comprises samples of five particles evolving within a 2D box without external forces (aside from possible wall collisions). Particle trajectories are simulated by solving two distinct partial differential equations (PDEs), one for each interaction type. Simulations run for 6000 time steps, with data subsampled every 100 steps. Each particle is represented by four features: position and velocity in both $x$ and $y$ directions. To simulate irregular partial observations, we adopt the approach from (Huang et al., 2020). For each particle, the number of observed points $n$ is drawn from a uniform distribution $U(40, 52)$, and $n$ time steps are uniformly sampled from the 6000-step trajectory for training. For extrapolation evaluation, 40 observations are sampled from steps [6000, 12000] using the same method. Sampling is performed independently for each particle. Following (Huang et al., 2020), we generate 20k training and 5k test samples for both datasets. All features are standardized to have a maximum absolute value of 1 in both the training and test sets.

*Table 7.* Statistics of the datasets used in the experiments.

|  | Spring | Charged | Motion-walk | Motion-jump | PEMS08 |
|---|---|---|---|---|---|
| #Nodes | 5 | 5 | 29 | 29 | 170 |
| #Node ft | 4 | 4 | 6 | 6 | 3 |
| #Edges | 1-20 | 25 | 56 | 56 | 548 |
| #Edge ft | 1 | 2 | 1 | 1 | 1 |
| #Train | 20000 | 20000 | 16 | 21 | 199 |
| #Test | 5000 | 5000 | 7 | 9 | 49 |
| #Number of samples in Train | $U(40,52)$ | $U(40,52)$ | $U(30,42)$ | $U(30,42)$ | $U(40,52)$ |
| #Number of samples in Test | $U(40,52)+40$ | $U(40,52)+40$ | $U(30,42)+40$ | $U(30,42)+40$ | $U(40,52)+40$ |

### D.2. CMU Datasets

The CMU dataset (CMU, 2003) is used for human pose recognition and motion analysis, containing three-dimensional human motion data obtained through motion capture systems such as optical cameras or sensors. We selected two datasets: the walk sequence of subject 35 and the jump sequence of subject 118. Each sample consists of 29 trajectories, each tracking a joint. Each joint has six features: displacement along the x, y, and z axes, and angles relative to the x, y, and z axes. Similar to the simulated datasets, for each joint in the motion capture data, we sample the number of observations $n$ from a uniform distribution $U(30, 42)$, and uniformly extract $n$ observations from the first 50 frames as the training trajectory. For testing, 40 observations are sampled from frames [51,99]. We divide the walk experiments into non-overlapping training (16 trials) and test sets (7 trials), and similarly, the jump experiments into non-overlapping training (21 trials) and test sets (9 trials). All features are normalized such that the maximum absolute value is 1 in both the training and testing datasets.

### D.3. PEMS08

The PEMS08 dataset (Song et al., 2020) originates from the traffic flow monitoring system in California, USA and records traffic flow data from various sensor points over different time periods. The PEMS08 dataset includes traffic flow data from 170 nodes over a continuous 62-day period, starting from July 1,2016. The data collection frequency is every 5 minutes, with each sample containing three features: the number of vehicles passing through in 5 minutes, the speed of the vehicles, and the occupancy rate. Similar to the generated datasets, for the training data, we divide the first 11,940 time steps into groups of 60 time steps, resulting in 199 non-overlapping subsequences. Observations for each node are randomly sampled from a uniform distribution $U(40, 52)$, with n observations selected from each group of 60 time steps. For the test set, data from time steps [11,941,17,820] is selected, and every 120 time steps are divided into a group, generating 49 non-overlapping subsequences. For each test sequence, the first 60 time steps are sampled using the same method as the training set, and 40 observations are randomly selected from the 61st to the 120th time steps. All features are standardized, ensuring that the maximum absolute value in both the training and test datasets is 1.

The statistics of the five datasets are reported in Table 7.

## E. Experiment Setup

### E.1. Interpolation

Fora system consisting of multiple nodes, the first observable time point for each node may differ. This means that the observation timestamps for each node are not aligned, and thus we cannot simply use a fixed time point as the starting time for all nodes. To handle these misaligned observation times consistently, we define a unified starting time $t_{\text{start}} = 0$ for all nodes, which serves as the initial time point for the system, and normalize all observation times to the range [0, 1]. This normalization process ensures that the time series for all objects are processed within a unified time frame, eliminating the issue of inconsistent time points across objects. By applying such time normalization, we assume that interactions between objects are continuous, starting from $t_{\text{start}} = 0$ and modeled within the time range [0,1]. This allows us to apply ODE within this standardized time frame to describe the dynamic behavior of the system and infer the complete trajectories based on the qiven partial observation data.

*Table 8.* Hyperparameter settings for every dataset .

| Hyper parameter | Values | | | | |
|---|---|---|---|---|---|
| | Spring | Charged | Motion-walk | Motion-jump | PEMS08 |
| batch | 256 | 256 | 8 | 32 | 4 |
| learing rate | | | 0.0005 | | |
| dropout | | | 0.2 | | |
| $k$ | 64 | 64 | 64 | 64 | 16 |
| $q$ | 32 | 32 | 32 | 32 | 8 |
| Alpha | | | 0.5 | | |
| M | | | 2 | | |
| clip | | | 10 | | |
| epoch | | | 50 | | |
| $h$ | | | 16 | | |
| $augment\_dim$ | 64 | 64 | 64 | 64 | 0 |
| L2 | | | 0.001 | | |
| subnetworks_width | 128 | 128 | 128 | 128 | 64 |
| subnetworks_dipth | | | 1 | | |
| ODEsolve | | | Euler | | |
| Optimizer | | | Adam | | |

## E.2. Extrapolation

During the training process, we manually segment each sequence, selecting the system's starting time as $t_{\text{start}} = t_{n1} = \frac{t_n - t_0}{2}$ ,and the observations prior to $t_{\text{start}}$ are fed into the encoder to estimate the latent initial state. Observations at or after $t_{\text{start}}$ are treated as ground truth data for reconstructing the latter part of the trajectory. For the testing process, in addition to using the first half of the data from $(t_0, t_n)$,we also sample observation data from the range $(t_n, t_{n2})$ to evaluate the model's extrapolation capability. Therefore, we set $t_{\text{start}} = t_{n'}$ with observations before $t_n$ being input into the encoder and observations at or after $t_n$ used as ground truth data to reconstruct the latter part of the trajectory. All observation times are normalized to the range [0, 1].

## E.3. Hyper-Parameter

In the Table 8, we report the hyperparameters used for all datasets in the experiments.

## F. Details of Baselines

We first consider Latent-ODE (Rubanova et al., 2019), which is suitable for modeling a single incomplete time series. This method uses NODE to encode information from incompletely sampled time points, but it does not account for interactions between nodes, making it incapable of directly capturing information transfer between nodes. Edge-GNN (Gong & Cheng, 2019) models time intervals as edge attributes, combining temporal information with graph structure to learn interactions between nodes via GNN. The time intervals are reflected in the edge attributes through a weighting mechanism, thus propagating temporal dynamic information across the graph. Weight-Decay (Cao et al., 2018) employs a simple exponential decay function to model time intervals as $h(t + \Delta t) = \exp\{-\tau\Delta t\} \cdot h(t)$,capturing the influence of time intervals on features through the decay function. However, it does not fully consider the dynamid interactions between nodes, as it is merely a simple parametrization of time. RNN-D (Che et al., 2018) is a multivariate time series (MTS) model based on RNNs for handling incomplete data. It fills missing values by connecting the feature vectors of all nodes uniformly, but it neglects interactions between nodes and assumes independence across all nodes, thus failing to model dynamic interactions and capture information transfer between nodes. NRI (Kipf et al., 2018) is a method for modeling relationships between nodes and predicting time series in dynamic systems, particularly suited for data with a graph structure. (Huang et al., 2020) combines RNN-D with NRI to form RNN-NRI, which is evaluated in two tasks. LG-ODE (Huang et al., 2020) combines the interaction functions of GNN with NODE, providing a flexible and powerful tool for modeling dynamic systems. It excels in continuous-time dynamic modeling, handling irregular sampling, and utilizing node interactions, making it suitable for a wide range of dynamic graph tasks. Additionally, we replace the ODE function in LG-ODE with the ODE function we

designed, retaining the encoding scheme of LG-ODE. This model is referred to as LG-CSODE, LG-CSODE is better suited for complex node evolution, offering significant advantages in capturing the nonlinear evolution of node states.

# G. Results Visualization

The Figure 5 visualizes the results of our model in the interpolation task on the Spring dataset, with observation percentages of 40% and 60%. The Figure 6 illustrates the results of our model in the extrapolation task on the Spring dataset

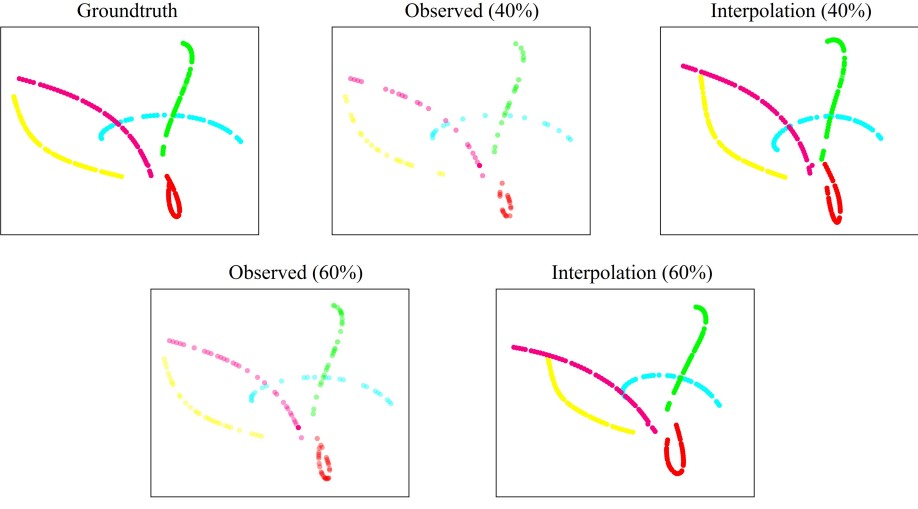

*Figure 5.* Visualization of interpolation results for spring system.

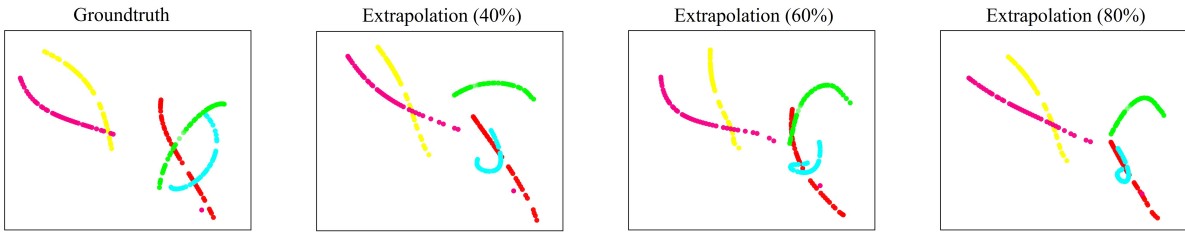

*Figure 6.* Visualization of extrapolation results for spring system.

# H. More Experiment Results

### H.1. Model Scaling

In the previous experiments, our model demonstrated significant advantages over various baseline models while maintaining the same number of parameters and architecture configuration. Building on these findings, our extended experiments primarily focus on exploring the scalability of the model, rather than further comparisons with other models Taking the walk capture data from Subject 35 as an example, we investigate the impact of increasing the number of sub-networks, sub-network width, and sub-network depth on system performance. Specifically, the network width refers to the number of hidden layers in the fully connected layers, set to 128, 256, 512, 1024, and 2048; the network depth refers to the number of hidden layers in the fully connected layers, set to 1,2,3,4,and 5; and the number of sub-networks is set to 1,2,3,4,and 5.

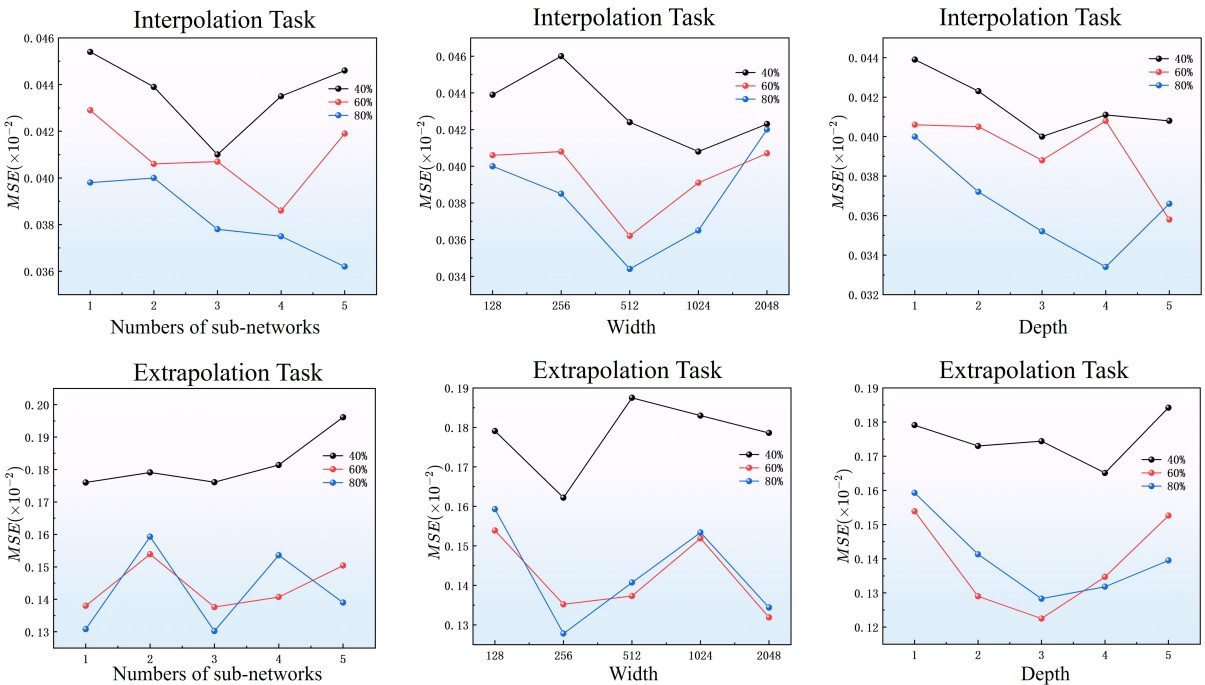

*Figure 7.* Comparison of the performance of the CSG-ODE model that increases only the number of sub-networks, sub-network width, or sub-network depth in both tasks (with fixed learning rate).

To maintain consistency across experiments, when adjusting one variable, the other two variables and the model's other hyperparameters (e.g., learning rate) remain unchanged.

In this experiment, we controlled the learning rate to be 0.0005, and the experimental results are shown in the Figure 7. In terms of overall performance, increasing the network width network depth, and number of sub-networks resulted in improved performance for both tasks. Moreover, we found that the MSE for both tasks first decreased and then increased as the three parameters above were increased, with the minimum MSE typically occurring whem these parameters were adjusted to their median values. For the CSODE ([Mei et al., 2024](#)) model, the learning rate formula used in the design of this experiment is: learning rate $= \frac{k}{width \times \sqrt{M}}$, where $k$ is a constant, and $M$ is the number of sub-networks. This formula adjusts the learning rate based on the width and moderately adjusts it according to the number of sub-networks to address the complexity. For example, with a width of 512 and 2 sub-networks, the learning rate is $\frac{1}{512 \times \sqrt{2}}$. Therefore, we applied this learning rate formula during the validation process to adjust the initial learning rate and observe the model's performance as complexity increases. Using the same data, we conducted experiments for both tasks with a 60% observation ratio, and the experimental results are shown in the Figure 8. The results indicate that increasing the network width, the number of sub-networks, and appropriately adjusting the learning rate can lead to a stable performance improvement for our model.

### H.2. Impact of Solver Choice

*Table 9.* Comparison of Mean Squared Error (MSE$\times 10^{-2}$) and running time of Euler and Dopri5 solvers for two tasks

| | Interpolation | | | | | | Extrapolation | | | | | |
|---|---|---|---|---|---|---|---|---|---|---|---|---|
| | 40% | Time | 60% | Time | 80% | Time | 40% | Time | 60% | Time | 80% | Time |
| Dopri5 | 0.0427 | 1600s | 0.0384 | 1700s | 0.0377 | 1800s | 0.1583 | 900s | 0.0919 | 900s | 0.0972 | 1000s |
| Euler | 0.0439 | 900s | 0.0406 | 950s | 0.0400 | 1050s | 0.1791 | 550s | 0.1539 | 550s | 0.1593 | 600s |

Differential equations are fundamental tools for describing many dynamic systems, and NODE use these equations to model

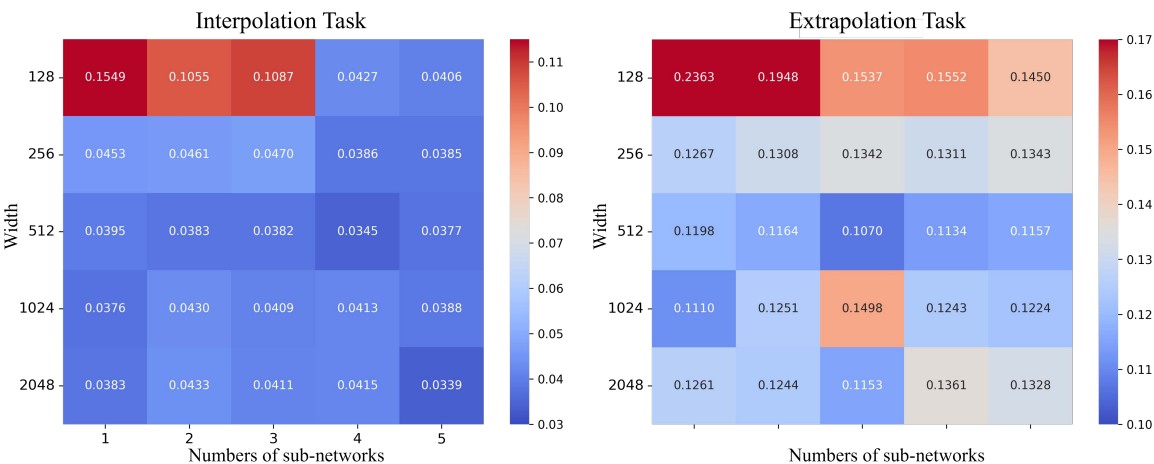

*Figure 8.* Comparison of the performance of CSG-ODE models with different numbers and widths of subnetworks on both tasks (dynamic adjustment of learning rate).

the dynamic evolution of nodes. In order to simulate the solution of differential equations via computers, numerical solvers are required to approximate the solutions of these equations. The choice of numerical solver directly influences both the accuracy of the model and its computational efficiency.

Numerical solvers can generally be classified into two categories: explicit methods and implicit methods. Explicit methods are relatively simple in terms of computation, as each step only depends on the current known values. As a result, they are usually faster in computation, but their accuracy may be lower. Implicit methods, on the other hand, involve not only the current known values but also require consideration of future values in each step. While this makes the computation more complex, implicit methods typically offer higher accuracy and better stability. Furthermore, solvers can also be classified based on their order of accuracy and their ability to adapt the step size. A higher order generally corresponds to better accuracy, while an adaptive step size allows the solver to dynamically adjust the step length based on error estimates, thereby improving computational efficiency without compromising precision.

The Euler method is a first-order explicit solver known for its simplicity and computational efficiency. Although it has lower accuracy, its simplicity makes it suitable for preliminary trials and rapid computations. Dopri5, a fifth-order explicit adaptive solver (a variant of the Runge-Kutta method), offers higher accuracy and can adjust the step size based on estimated errors, thus providing a balance between accuracy and computation time. In this experiment, we used the walk capture data from Subject 35 as an example to compare the impact of two solvers (Euler's method and Dopri5) on the model's prediction accuracy and computational efficiency.

*Table 10.* Friedman test statistics

| Task | ratio | Friedman statistic($\chi^2$) | Degrees of Freedom | Significance $p_s$ |
|---|---|---|---|---|
| | 40% | 28.54 | 6 | $7.42\times10^{-5}$ |
| Interpolation | 60% | 28.97 | 6 | $6.16\times10^{-5}$ |
| | 80% | 28.03 | 6 | $9.28\times10^{-5}$ |
| | 40% | 25.80 | 6 | $2.42\times10^{-4}$ |
| Extrapolation | 60% | 26.40 | 6 | $1.88\times10^{-4}$ |
| | 80% | 27.77 | 6 | $1.04\times10^{-4}$ |

The results are presented in the Table 9. The use of the Dopri5 method significantly improved the model's performance, particularly in the extrapolation task, where a substantial enhancement in performance was observed. However, compared to the Euler method, the computational time for Dopri5 increased on average by 71.7%.

### H.3. Statistical Significance Analyses

In order to verify the robustness of the experimental results and to strengthen the empirical claims, we conducted the Friedman test on the experimental results in Table 1 and Table 2. The statistical results of the Friedman test are shown in Table 10. Among them, the significance level $\alpha_s = 0.05$, the significance $p_s$ are much less than the significant level, which indicates the rejection of the original hypothesis (the original hypothesis is that there is no significant difference in the predictive effect of these seven models), i.e., it indicates that there is a significant difference in the predictive effect of the seven models on the individual datasets. And the larger the Friedman statistic ($\chi^2$), the more significant the difference in prediction results within the set.

## I. Algorithm

We summarize the learning algorithm of our CSG-ODE in Algorithm 1.

---

**Algorithm 1** Training Algorithm of CSG-ODE

---

    **Input:** Observation data $\mathcal{G}^{T_1:T_{obs}}$
    **Output:** The parameters in the model
    Initialize model parameters
    **while** not convergence **do**
        **for** each training sequence **do**
            Construct the temporal graph with Equation (5)
            Generate a representation of each node by Equation (7)
            Generate an approximate posterior distribution of potential states for each node using Equation (12)
            Sample the initial latent state $z_i^0$ of each node
            Solve our ODE in Equation (14)
            Output the trajectories using the decoder
            Compute the final objective, i.e., Equation (17)
            Update parameters in our CSG-ODE using gradient descent
        **end for**
    **end while**

---

