# OpenReview forum: "CSG-ODE: ControlSynth Graph ODE For Modeling Complex Evolution of Dynamic Graphs"
_ICML.cc/2025/Conference — ICML 2025 poster_

### Official Review · Reviewer_xUJZ · 2025-03-09

**Overall Recommendation:** 3

**Summary:**

The paper proposes a new approach called CSG-ODE for modeling the evolution of dynamic graphs. The main contribution lies in introducing an information transmission-based inter-node importance weighting mechanism and utilizing nonlinear activation functions in the ODE-based modeling. The authors claim that this approach improves the stability and performance of dynamic graph models, particularly in traffic, motion capture, and simulated physical systems. The paper also presents an extension of CSG-ODE, termed Stable CSG-ODE, which theoretically guarantees enhanced stability.

**Claims And Evidence:**

The provided evidence is mostly convincing, with the authors demonstrating strong experimental results comparing CSG-ODE against baseline models. However, the claims about stability improvements should be further supported by analyzing the model's performance under specific conditions. In which concrete scenarios might this method exhibit relatively better stability?

**Essential References Not Discussed:**

The authors do not discuss some recent works on graph neural networks for temporal and dynamic systems that could provide important context for their contributions. For example, the work by Yıldız et al. (2022) on learning interacting dynamical systems with latent Gaussian process ODEs offers a related approach that could inform their model’s latent space learning.

**Experimental Designs Or Analyses:**

The experimental design is sound, with a well-chosen set of datasets and baseline models for comparison. The paper does a good job of demonstrating the effectiveness of CSG-ODE and SCSG-ODE in real-world applications. However, the analysis could be more comprehensive in certain areas. For example, while the authors mention the use of a control function for modeling node interactions, they do not provide sufficient insight into how this control information interacts with other parts of the model. Additionally, a more detailed error analysis for each dataset would help clarify the strengths and weaknesses of the proposed method.

**Methods And Evaluation Criteria:**

The proposed method is sound and appropriately chosen for the problem of dynamic graph modeling.
In the experimental results presented in Appendix Table 5, the proposed methods SCSG-ODE and CSG-ODE demonstrate advantages over other baselines, but the stability advantages of SCSG-ODE are not directly evident from these results.

**Other Comments Or Suggestions:**

The appendix of the paper contains a lot of content, some of which is redundant. Some important experimental results in the appendix should be placed in the main body as much as possible, and some content needs to be condensed or removed. For example, the relevant introduction of the dataset and baseline methods in the appendix are quite redundant and need to be expressed concisely. Also, the introduction about GNN is actually not necessary.

In addition, the abstract mentions "For high-stability scenarios...", but the content of the paper refers to "To enhance stability in high-stability scenarios" or "the demands of high-stability scenarios," so the expression in the abstract is problematic.

**Other Strengths And Weaknesses:**

One strength of the paper lies in its integration of node importance weighting and nonlinear activation functions to capture complex node dynamics. The theoretical foundation for the model's stability represents a significant contribution to the literature, and the experimental results are relatively compelling. However, a minor weakness is that while the authors claim their model delivers a more stable solution, the stability experiments in the appendix are not very convincing.

**Questions For Authors:**

The training procedure of the model lacks detailed description, particularly regarding how hyperparameters (such as weight matrices) are tuned during training. Are there any specific guidelines or best practices for selecting these hyperparameters?

**Relation To Broader Scientific Literature:**

The authors place their work in the context of ODE-based models for dynamic graph learning, which have gained traction in recent years. The contribution of CSG-ODE lies in its combination of latent space modeling and graph-based learning. The addition of the inter-node importance weighting mechanism further differentiates it from previous methods like LatentODE and GODE. Moreover, the extension to SCSG-ODE contributes to the literature by addressing issues of stability, which have been underexplored in prior works.

**Theoretical Claims:**

The theoretical aspects of the paper, particularly the stability guarantee provided by SCSG-ODE, are interesting and well-supported. However, some of the notation and mathematical details could benefit from additional clarity. For instance, the explanation of the learnable anti-symmetric weight matrices in SCSG-ODE is not fully accessible to readers who may not be familiar with the underlying mathematical principles. A more intuitive explanation of why this form enhances stability would help strengthen the theoretical contribution.

---

> ### Author Rebuttal · Authors · 2025-03-31
>
> **Thank you for your valuable comments. We have responded to each concern as follows:**
>
> A1(Response to Claims And Evidence):
>
> Suggestion(1): To verify the correctness of the theoretical derivations, we perform additional extrapolation experiments on CSG-ODE and SCSG-ODE on walk motion data. Each model was run for 5 rounds and its mean and standard deviation were calculated $(MSE \times{10^{-2}})$. The results show that the standard deviation of SCSG-ODE is always about half of that of CSG-ODE under all sampling ratios, which indicates that it has higher stability, thus confirming the correctness of the theoretical derivation. The experimental results are shown in the following table:
>
> ratio 40% 60% 80%
>
> CSG-ODE $0.1883\pm0.0092$ $0.1676\pm0.0109$ $0.1524\pm0.0097$
>
> SCSG-ODE $0.2304\pm0.0056$ $0.1978\pm0.0050$ $0.1787\pm0.0043$
>
> Question(2): We believe that it depends on the demand for the task, and in experiments where stability is important, a tiny fraction of the model performance can be sacrificed in favor of SCSG-ODE.
>
> A2(Response to Methods And Evaluation Criteria):
>
> See A1.
>
> A3(Response to Theoretical Claims):
>
> Definition of antisymmetric matrix: The matrix $A$ satisfies ${A^T} = - A$, i.e., $a_{ij}=-a_{ji}\forall i,j$. For any matrix $A$, it can be converted to an antisymmetric matrix by $A-A^T$ (given in Remark 3.1 in the text). In our experiments, we learn a matrix $A$ and ensure its antisymmetry with the help of methods in Remark 3.1. Antisymmetric matrices have important mathematical properties and physical significance in linear algebra, and their purely imaginary eigenvalue property ensures the stability of ODE systems.
>
> A4(Response to Experimental Designs Or Analyses):
>
> Question(1): We model the node state evolution using a dynamical system control framework (Eqn.(18)), where ${A_0}z_i^t$ describes the linear dynamics, and $c_i^t$ (the neighborhood interaction information computed by the GNN) is transformed into control inputs (functionally analogous to the control matrix $B$ in the classical control) by the neural network $g(\cdot)$. Different from the classical control theory, we introduce subnetworks with nonlinear activation functions to portray the nonlinear evolution of the node states, so that the node states are driven by a combination of linear evolution, nonlinear evolution and external control.
>
> Question(2): We performed a more detailed error analysis of the model for each dataset. For real datasets, numerical errors in ODE solving are inevitable. In addition, real-world data are often affected by complex factors, such as environmental changes, that were not included in our consideration and may further exacerbate error accumulation.
>
> A5(Response to Supplementary Material):
>
> We have added the training algorithm:
>
> \STATE {\bfseries Input:}Observation data
>
> \STATE {\bfseries Output:}The parameters in the model
>
> \STATE Initialize model parameters
>
> \WHILE {not convergence}
>
> \FOR {each training sequence}
>
> \STATE Construct the temporal graph with Eqn. 5
>
> \STATE Generate a representation of each node by Eqn. 8 to 11
>
> \STATE Generate an approximate posterior distribution of potential states for each node using Eqn.16
>
> \STATE Sample the initial latent state $z_i^0$ of each node
>
> \STATE Solve our ODE in Eqn. 18
>
> \STATE Output the trajectories using the decoder
>
> \STATE Compute the final objective, Eqn. 21
>
> \STATE Update parameters in our CSG-ODE using gradient descent
>
> \ENDFOR
>
> \ENDWHILE
>
> A6(Response to Relation To Broader Scientific Literature):
>
> We add the following discussion: Yıldız et al. (2022) accurately decompose the independent dynamics of individual objects from their interactions and infer the independent dynamics and its interaction with reliable uncertainty estimates using ordinary differential equations for underlying Gaussian processes.
>
> A7(Response to Relation To Other Strengths And Weaknesses):
>
> See A1.
>
> A8(Response to Other Comments Or Suggestions):
>
> Suggestion(1):We put the results on the SCSG-ODE into the main text, removing the description of the GNN in the appendix, while simplifying the description of the dataset and the baseline methodology.
>
> Suggestion(2): We have changed the abstract to read that for scenarios with stability requirements or prediction tasks...
>
> A9(Response to Questions For Authors):
>
> Question(1): See A5.
>
> Question(2): $\beta$ denotes the step size of the finite difference approximation, and we use the empirical Eqn.$\beta = \frac{2}{N} \times {10^{ - 4}}$ proposed by Noschese(2024) for $\beta$ selection, which is based on an error analysis of finite-difference approximation and balances the truncation and rounding errors to ensure that the total error is minimized. For $\alpha$, we performed a sensitivity analysis (see Section 4.6). We finally chose to conduct most of the experiments with $\alpha=0.5$ because in the extrapolation experiments, the results around $\alpha=0.5$ tend to be stable and locally optimal.

---

### Official Review · Reviewer_QZ2k · 2025-03-10

**Overall Recommendation:** 4

**Summary:**

The paper introduces the ControlSynth Graph (CSG-ODE) model, which improves iupon existing Graph Neural ODE models for dynamic graph representation. The model incorporates node importance weights based on information propagation and employs multiple subnetworks with nonlinear activation functions to better capture the nonlinear evolution of node states. Additionally, the paper presents an extension, Stable CSG-ODE (SCSG-ODE), which theoretically improves model stability.

Extensive experimental evaluations on several dynamic system datasets demonstrate that CSG-ODE outperforms existing models, while SCSG-ODE excels in both stability and performance, particularly in high-stability scenarios.

**Claims And Evidence:**

Yes.

The claims regarding the stability improvements in SCSG-ODE are theoretically supported.

The experimental validation of these stability improvements could be more explicitly demonstrated.

**Essential References Not Discussed:**

n/a

**Experimental Designs Or Analyses:**

yes.

However, the current experimental results lack statistical significance tests, raising concerns about whether the proposed method is statistically superior to other approaches. Including statistical significance analyses (e.g., confidence intervals, hypothesis testing, or variance analysis) would help validate the robustness of the results and strengthen the empirical claims.

**Methods And Evaluation Criteria:**

The proposed methods and evaluation criteria are well-suited to the problem.

However, one concern is how the node importance mechanism specifically enhances performance compared to models that rely solely on latent node representations. For instance, in dynamic graphs with complex node relationships, does incorporating node importance weighting lead to a significant performance improvement?

**Other Comments Or Suggestions:**

Some of the mathematical notation could be clarified, such as the information propagation-based inter-node importance weighting mechanism.

Additionally, a clearer description of the hyperparameter tuning process, especially regarding stability-related parameters, would improve the reproducibility and interpretability of the model.

**Other Strengths And Weaknesses:**

Strengths:

	1.	The introduction of node importance weighting and nonlinear dynamics in ODE-based graph modeling is novel and promising, offering a more expressive approach to dynamic graph representation.

	2.	The experimental results are convincing, demonstrating that CSG-ODE outperforms existing methods across multiple datasets, supporting the model’s effectiveness.

	3.	The theoretical contribution regarding stability in SCSG-ODE provides a valuable addition to the literature, offering insights into stability improvements in dynamic graph ODEs.

Weaknesses:

	1.	While the stability of SCSG-ODE is theoretically proven, the experimental validation of stability improvements remains insufficient, requiring more empirical evidence to support the theoretical claims.

	2.	Certain aspects of the experimental analysis, such as error analysis and the impact of individual model components, could be more detailed to provide a deeper understanding of the model’s behavior and limitations.

**Questions For Authors:**

What are the potential limitations of the SCSG-ODE model in terms of scalability to very large dynamic graphs, and how might these limitations be addressed in future work?


## update after rebuttal

Thank you for the detailed rebuttal, which addressed my main concerns. I am pleased to recommend Accept.

**Relation To Broader Scientific Literature:**

By introducing adaptive node importance and nonlinear ODEs, this paper improves upon existing methods that primarily rely on linear dynamics. These enhancements enable the model to capture more complex temporal dependencies and better represent the evolving structures of dynamic graphs, addressing limitations of traditional approaches.

**Theoretical Claims:**

yes.

The theoretical claims regarding the stability of SCSG-ODE are supported by a theoretical proof provided in the appendix.

---

> ### Author Rebuttal · Authors · 2025-03-31
>
> **Thank you for your valuable comments. We have responded to each concern as follows:**
>
> A1(Response to Methods And Evaluation Criteria):
>
> The mechanism improves model performance in the following ways:
> - Make up for the lack of local information: by measuring the contribution of each edge to the total information propagation in the graph, we make the weight not only contain the original topological information, but also portray the influence of edges on the overall information flow, thus enhancing the model's ability to utilize the global information.
> - Explanation of real-world application: in a transportation network, the efficiency of information flow will be improved if the accessibility between nodes $i$ and $j$ is enhanced (e.g., by widening the lanes). Our approach enables the model to more accurately simulate dynamic interactions in complex networks by measuring the role of edges in overall information propagation.
> - Experimental Validation: The results of the ablation experiments show a significant decrease in model performance after the remove of this module, further demonstrating the key role of this mechanism in improving the effectiveness of the model.
>
> A2(Response to Experimental Designs Or Analyses):
>
> We performed Friedman's test (significance level $\alpha_s = 0.05$):
>
> Task ratio Friedman statistic
>
> Interpolation 40% 28.54
>
> ~ 60% 28.97
>
> ~ 80\% 28.03
>
> Extrapolation 40% 25.80
>
> ~ 60% 26.40
>
> ~ 80% 27.77
>
> The critical value of the Friedman test is 2.508 when $k_s = 7,\alpha_s = 0.05$, and $N_s = 5$, where $k_s$ denotes the number of models compared and $N_s$ denotes the number of data sets. Since the calculated Friedman statistic is much larger than the critical value, there is a significant difference in the predictive performance among the seven models. In addition, the larger the Friedman statistic, the more significant the difference in prediction results.
>
> A3(Response to Supplementary Material):
>
> To verify the correctness of the theoretical derivations, we perform additional extrapolation experiments on CSG-ODE and SCSG-ODE on walk motion data. Each model was run for 5 rounds and its mean and standard deviation were calculated $(MSE \times{10^{-2}})$. The results show that the standard deviation of SCSG-ODE is always about half of that of CSG-ODE under all sampling ratios, which indicates that it has higher stability, thus confirming the correctness of the theoretical derivation. The experimental results are shown in the following table:
>
> ratio 40% 60% 80%
>
> CSG-ODE $0.1883\pm0.0092$ $0.1676\pm0.0109$ $0.1524\pm0.0097$
>
> SCSG-ODE $0.2304\pm0.0056$ $0.1978\pm0.0050$ $0.1787\pm0.0043$
>
> A4(Response to Relation To Other Strengths And Weaknesses):
>
> Weaknesses(1): See A3.
>
> Weaknesses(2): We performed a more detailed error analysis for each dataset. For real datasets, numerical errors in ODE solving are unavoidable. In addition, real-world data are affected by complex factors such as environmental changes that are not taken into account and may exacerbate the accumulation of errors. In the ablation experiments, we further analyze the effects of each component. Among them, the Ours-no EI performance is significantly reduced because the weights not only encode the topology of the original graph, but also reflect the role of edges in the global information propagation, which improves the model's ability to portray time-varying relationships among nodes.
>
> A5(Response to Other Comments Or Suggestions):
>
> Suggestion(1):${L_f}(G_o^T,e{e^T})$ is the Frechet derivative with regard to $G_o^T$ and $e{e^T}$, which denotes the total transmission rate, and is used to measure the contribution of each edge in the graph to the information transfer. $|| \cdot |{|_F}$ is the Frobenius norm,$||{L_f}(G_o^T,e{e^T})|{|_F}$ denoting the total transmissibility of the graph. ${L_f}(G_o^T,e{e^T})$ is computed by finite difference approximation, where $\beta$ denotes the step size.
>
> Suggestion(2): We use the empirical Eqn.$\beta = \frac{2}{N} \times {10^{ - 4}}$ proposed by Noschese(2024) for $\beta$ selection, which is based on an error analysis of finite-difference approximation and balances the truncation and rounding errors to ensure that the total error is minimized. For $\alpha$, we performed a sensitivity analysis (see Section 4.6). We finally chose to conduct most of the experiments with $\alpha=0.5$ because in the extrapolation experiments, the results around $\alpha=0.5$ tend to be stable and locally optimal.
>
> A6(Response to Questions For Authors):
>
> In large dynamic graphs, it may be difficult to adequately capture the complex interaction patterns between nodes by relying only on GNN as graph information aggregators. Therefore, we plan to construct modeling methods that are more in line with real information propagation in order to model dynamic interaction processes more effectively. In addition, we plan to incorporate external environmental variables to enhance the generalization ability of the model.

---

### Official Review · Reviewer_1rWV · 2025-03-10

**Overall Recommendation:** 4

**Summary:**

This paper focuses on graph ODE model that handles dynamic relations and nodes with non-linear state evolution. The paper proposes a model called CSG-ODE that incorporates learnable latent graphs and time-varied graph snapshots. The CSG-ODE involves multiple dynamic subgraphs to capture state change of nodes. Experimental results on different datasets validate the effectiveness of CSG-ODE.

**Claims And Evidence:**

The proposed CSG-ODE is claimed to be aiming at capturing time-varying relationships. However, there is no statistical analyses or cases illustrating such characteristics. Additionally, the SCSG-ODE has been proven to represent stable dynamic systems, yet no experimental metric is given to evaluate the stability of different ODEs.

**Essential References Not Discussed:**

NA.

**Experimental Designs Or Analyses:**

The experiment and analyses are conducted properly.

**Methods And Evaluation Criteria:**

The evaluation criteria follows the common practice of researches on graph ODEs.

**Other Comments Or Suggestions:**

NA.

**Other Strengths And Weaknesses:**

Strengths:

+ The proposed framework incoporates the external control signals in dynamic systems, which is a valuable and novel idea in graph ODEs.
+ The comprehensive experiments illustrate the effectiveness and functionalities of the proposed CSG-ODE.
+ Theoretical proofs and analyses are provided to support the stability of the  SCSG-ODE model.

**Questions For Authors:**

1. It seems that the control $c_i^t$ is updated via ODEs that fully contolled by $z_t$. Could you explain how the $c_i^t$ capture the external information of the dynamic systems?

**Relation To Broader Scientific Literature:**

The proposed CSG-ODE enhances existing graph ODEs with extra considerations regarding the dynamic and implicit relationship between nodes.

**Theoretical Claims:**

I have gone through the theoretical analysis.

---

> ### Author Rebuttal · Authors · 2025-03-31
>
> **Thank you for your valuable comments. We have responded to each concern as follows:**
>
> A1(Response to Claims And Evidence):
>
> Suggestion(1): Information Propagation based Inter-node Importance Weight can better capture the time-varying relationship in the following aspects:
> - Make up for the lack of local information: by measuring the contribution of each edge to the total information propagation in the graph, we make the weight not only contain the original topological information, but also portray the influence of edges on the overall information flow, thus enhancing the model's ability to utilize the global information.
> - Explanation of real-world application: in a transportation network, the efficiency of information flow will be improved if the accessibility between nodes $i$ and $j$ is enhanced (e.g., by widening the lanes). Our approach enables the model to more accurately simulate dynamic interactions in complex networks by measuring the role of edges in overall information propagation.
> - Experimental Validation: In the ablation experiment, we remove the module. The results show that the model performance decreases significantly after removing the module, which further proves the key role of the mechanism in improving the effectiveness of the model.
>
> Suggestion(2): To verify the correctness of the theoretical derivations, we perform additional extrapolation experiments on CSG-ODE and SCSG-ODE on walk motion data. Each model was run for 5 rounds and its mean and standard deviation were calculated $(MSE \times{10^{-2}})$. The results show that the standard deviation of SCSG-ODE is always about half of that of CSG-ODE under all sampling ratios, which indicates that it has higher stability, thus confirming the correctness of the theoretical derivation. The experimental results are shown in the following table:
>
> ratio 40% 60% 80%
>
> CSG-ODE $0.1883\pm0.0092$ $0.1676\pm0.0109$ $0.1524\pm0.0097$
>
> SCSG-ODE $0.2304\pm0.0056$ $0.1978\pm0.0050$ $0.1787\pm0.0043$
>
> A2(Response to Questions For Authors):
>
> The $c_i^t$ denotes the node interaction information computed by the GNN, which we consider as the external control information of the node, and is updated as shown in Eqn(18). The equation contains two ODE equations, where the second ODE equation describes the continuous evolution of interactions between nodes and other nodes. Similarly to the classical discrete GNN, we perform a continuum treatment that allows $c_i^t$ to be modeled as an ODE. Specifically, in this model, the control information $c_i^t$ is computed by the GNN, which acts as a graph information aggregator that influences the dynamic evolution of the node's own state by aggregating information from neighboring nodes and transforming it into continuous control signals.

---

### Official Review · Reviewer_9q63 · 2025-03-10

**Overall Recommendation:** 3

**Summary:**

The paper proposes a novel model CSG-ODE and its stable variant SCSG-ODE for continuous modeling of dynamic graphs. The approach integrates a VAE framework with neural ODEs, introducing an information propagation–based inter-node importance weighting and multiple nonlinear subnetworks to capture complex node state evolution. The authors validate their claims with extensive experiments on traffic, motion capture, and simulated physical systems, demonstrating improvements on both interpolation and extrapolation tasks compared to several baselines.

**Claims And Evidence:**

The paper makes several well-supported claims, notably regarding the improved modeling of time-varying node relationships and nonlinear state evolution, as evidenced by extensive experiments across multiple datasets. However, the specific impact of the β parameter on the information propagation weighting mechanism is not entirely clear. While the experimental results generally support the model’s effectiveness, additional quantitative analysis or sensitivity studies on β could provide a more comprehensive understanding.

**Essential References Not Discussed:**

The manuscript cites many key works; including a discussion of alternative approaches could further enrich the context. For instance, Temporal Graph Networks (TGN) by Rossi et al. (NeurIPS 2020) offer a discrete-time framework for dynamic graph representation that contrasts with the continuous-time approach adopted here. In addition, the work on Neural Controlled Differential Equations for Irregular Time Series by Kidger et al. (NeurIPS 2020) provides insights into handling irregular sampling, which is relevant to the challenges addressed in this paper.

**Experimental Designs Or Analyses:**

The experimental design is comprehensive, covering multiple datasets and tasks, and includes useful ablation studies and sensitivity analyses. The comparison with existing methods is thorough. One suggestion is to provide more detailed quantitative discussion on the effect of the sampling density adjustment mechanism to help readers better appreciate its role.

**Methods And Evaluation Criteria:**

The proposed methods are well-motivated and fit the problem domain. The combination of a VAE with neural ODEs to model irregularly sampled data is appropriate, and the use of MSE as the evaluation metric is standard for these tasks. However, some parts of the method, particularly the lengthy derivations and symbol definitions in the model formulation, could be streamlined for clarity.

**Other Comments Or Suggestions:**

I suggest that the authors simplify parts of the derivation and include more detailed explanations for specific parameters (e.g., the selection of β). Additionally, a brief discussion of the computational overhead introduced by the node importance module would be beneficial.

**Other Strengths And Weaknesses:**

Strengths:
•	The paper presents an innovative combination of node importance weighting with nonlinear ODE modeling. The integration of information propagation to adjust node interactions directly addresses limitations in existing approaches.

•	The experimental evaluation is comprehensive. Detailed ablation studies and comparisons across multiple datasets clearly demonstrate the contribution of each model component.

•	The stability proof, based on the properties of antisymmetric matrices and Jacobian eigenvalue analysis, provides concrete theoretical support for the model design.

Weaknesses:

•	The link between the theoretical stability proof and the observed empirical performance is not explicitly discussed.

•	Some areas need more clarity, such as adding brief explanations for intermediate steps in the proof (e.g., the Jacobian derivation J(t)=PA), while unnecessary basic content like "C. Detail of GNN" should be removed.

**Questions For Authors:**

•	What considerations led to the choice of the β parameter, and how does it quantitatively affect the information propagation weighting?

•	How does the non-negativity of the activation function’s derivative relate to matrix invertibility in your stability proof?

**Relation To Broader Scientific Literature:**

The paper positions itself well within the current literature on graph neural networks, neural ODEs, and VAE-based time series modeling. It builds on prior work such as Latent-ODE, LG-ODE, and NRI+RNN, and its contributions are clearly distinguished. Additionally, the authors could further elaborate on how their approach mitigates common issues in existing methods—such as the over-smoothing problem in graph neural ODEs and challenges in handling irregular sampling—by contrasting their solution with recent models that specifically address these challenges.

**Theoretical Claims:**

The paper offers a stability proof for the SCSG-ODE model based on the properties of antisymmetric matrices and Jacobian eigenvalue analysis. The proof is generally correct but would benefit from clearer explanations in parts—such as the relationship between the non-negativity of the activation function’s derivative and the matrix invertibility. Overall, the theoretical claims are sound but could be refined in presentation.

---

> ### Author Rebuttal · Authors · 2025-03-31
>
> **Thank you for your valuable comments. We have responded to each concern as follows:**
>
> A1(Response to Claims And Evidence):
>
> ${L_f}(G_o^T,e{e^T})$ is the Frechet derivative with regard to $G_o^T$ and $e{e^T}$, which denotes the total transmission rate, and is used to measure the contribution of each edge in the graph to the information transfer. $||\cdot|{|_F}$ is the Frobenius norm,$||{L_f}(G_o^T,e{e^T})|{|_F}$ denoting the total transmissibility of the graph. ${L_f}(G_o^T,e{e^T})$ is computed by finite difference approximation, where $\beta$ denotes the step size, and the shorter the step the more accurate it is. We use the empirical equation $\beta = \frac{2}{N} \times {10^{-4}}$ proposed by Noschese(2024) for $\beta$ selection, which is based on the error analysis of the finite-difference approximation, and is able to balance the truncation and rounding errors to ensure that the total error is minimized.
>
> A2(Response to Methods And Evaluation Criteria):
>
> We remove the well-known graph convolution Eqn (7) in our modification. We define more explicitly the mathematical symbol for Information Propagation based Inter-node Importance Weight, as described in A1.
>
> A3(Response to Theoretical Claims):
>
> A diagonal matrix $P$ is invertible if none of its diagonal elements are zero, and by the nonnegativity of the activation function we can obtain that none of the diagonal elements of the matrix $P$ is zero, and therefore the matrix $P$ is invertible. We add this in the text.
>
> A4(Response to Experimental Designs Or Analyses):
>
> $\alpha$ is a key parameter for adjusting the effect of the sampling density mechanism, and we performed sensitivity experiments and analyzed it (see Section 4.6).
>
> A5(Response to Supplementary Material):
>
> Given a vector-valued function $f:\mathbb{R}^n\to\mathbb{R}^m$, its Jacobian matrix ${J_f}(x)$ is a $m \times n$ matrix, with each element denoting the partial derivative of $f$ with respect to the input variable: $J_f(x)=\begin{bmatrix}\frac{\partial f_1}{\partial x_1}& \frac{\partial f_1}{\partial x_2}& \cdots &\frac{\partial f_1}{\partial x_n}\\\ \frac{\partial f_2}{\partial x_1}&\frac{\partial f_2}{\partial x_2}&\cdots&\frac{\partial f_2}{\partial x_n}\\\ \vdots&\vdots&\ddots&\vdots\\\ \frac{\partial f_m}{\partial x_1}&\frac{\partial f_m}{\partial x_2}&\cdots&\frac{\partial f_m}{\partial x_n}\end{bmatrix}$. We have added this intermediate step in the text. Also deleted the reference to GCN in Appendix C.
>
> A6(Response to Relation To Broader Scientific Literature):
>
> The graph neural ODE converts message propagation into differential equations, which can adaptively adjust the computational depth to avoid the over-smoothing problem of too deep GNN. Based on ODE solving, it can deal with arbitrary time intervals without predefined time step limitations, which is suitable for irregularly sampled data. Therefore, the graph neural ODE is inherently equipped with the advantages of mitigating the oversmoothing problem and dealing with irregular sampling.
>
> A7(Response to Relation To Other Strengths And Weaknesses):
>
> Weaknesses(1): To verify the correctness of the theoretical derivations, we perform additional extrapolation experiments on CSG-ODE and SCSG-ODE on walk motion data. Each model was run for 5 rounds and its mean and standard deviation were calculated $(MSE \times{10^{-2}})$. The results show that the standard deviation of SCSG-ODE is always about half of that of CSG-ODE under all sampling ratios, which indicates that it has higher stability, thus confirming the correctness of the theoretical derivation. The experimental results are shown in the following table:
>
> ratio 40% 60% 80%
>
> CSG-ODE $0.1883\pm0.0092$ $0.1676\pm0.0109$ $0.1524\pm0.0097$
>
> SCSG-ODE $0.2304\pm0.0056$ $0.1978\pm0.0050$ $0.1787\pm0.0043$
>
> Weaknesses(2):We provide a brief description of the Jacobian derivation $J(t)$, as described in A5. The contents of Appendix C have also been deleted.
>
> A8(Response to Other Comments Or Suggestions):
>
> Suggestion(1):We have added the following to the text for the answer in A1
>
> Suggestion(2):We have added the following to the text: Theorem: For the computation of the node importance weight matrix $D\in\mathbb{R}^{N\times N}$, the time complexity is $O({N^3})$.
> Note: where $\exp_0(G_o)$ is needed to compute the matrix indices by power series expansion, which usually requires $O({N^3})$ operations due to the matrix multiplication operations involved. The Frobenius norm and the linear operations that follow produce an additional $O({N^2})$ computation, but this is covered by the dominant term, $O({N^3})$.
>
> A9(Response to Questions For Authors):
>
> Question(1):See A1.
>
> Question(2):See A3.

---

### Decision · Program_Chairs · 2025-05-01

**Decision:**

Accept (poster)

**Comment:**

This paper received four effective reviews, and all of them are positive. Overall, the paper is of good quality and should be accepted.